# From "wading through treacle" to "making haste slowly": A comprehensive yet parsimonious model of drivers and challenges to implementing patient data sharing projects based on an EPaCCS evaluation and four pre-existing literature reviews

Mila Petrova[ID]*, Stephen Barclay[ID]

Palliative and End of Life Care Group in Cambridge (PELiCam), Primary Care Unit, Department of Public Health and Primary Care, University of Cambridge, United Kingdom

* mp686@medschl.cam.ac.uk

## Abstract

Conceptually, this study aimed to 1) identify the challenges and drivers encountered by England's Electronic Palliative Care Coordination System (EPaCCS) projects in the context of challenges and drivers in other projects on data sharing for individual care (also referred to as Health Information Exchange, HIE) and 2) organise them in a comprehensive yet parsimonious framework. The study also had a strong applied goal: to derive specific and non-trivial recommendations for advancing data sharing projects, particularly ones in early stages of development and implementation. Primary data comprised 40 in-depth interviews with 44 healthcare professionals, patients, carers, project team members and decision makers in Cambridgeshire, UK. Secondary data were extracted from four pre-existing literature reviews on Health Information Exchange and Health Information Technology implementation covering 135 studies. Thematic and framework analysis underpinned by "pluralist" coding were the main analytical approaches used. We reduced an initial set of >1,800 parameters into >500 challenges and >300 drivers to implementing EPaCCS and other data sharing projects. Less than a quarter of the 800+ parameters were associated primarily with the IT solution. These challenges and drivers were further condensed into an action-guiding, strategy-informing framework of nine types of "pure challenges", four types of "pure drivers", and nine types of "oppositional or ambivalent forces". The pure challenges draw parallels between patient data sharing and other broad and complex domains of sociotechnical or social practice. The pure drivers differ in how internal or external to the IT solution and project team they are, and thus in the level of control a project team has over them. The oppositional forces comprise pairs of challenges and drivers where the driver is a factor serving to resolve or counteract the challenge. The ambivalent forces are factors perceived simultaneously as a challenge and a driver depending on context, goals and perspective. The framework is distinctive in its emphasis on: 1) the form of challenges and drivers; 2) ambivalence, ambiguity and persistent tensions as fundamental forces in the field of innovation

**Data Availability Statement:** A significant subset of the interviews are with members of the project team and with individuals with leadership roles in the local health economy. Openly releasing the transcripts of such interviews can easily compromise participants' anonymity. Over 30,000 words of processed qualitative data are provided in S2 File and in the OSF (Open Science Framework) depository at https://osf.io/fkcag.

**Funding:** This paper presents independent research funded by the National Institute for Health Research (NIHR) Applied Research Collaboration (ARC) East of England, at Cambridgeshire and Peterborough NHS Foundation Trust; the Health Innovation and Education Cluster (HIEC) hosted by Cambridge University Health Partners (CUHP); and by The Marie Curie Design to Care programme. The Marie Curie Design to Care programme is a service improvement programme. This research forms part of the design phase of this programme, which is funded by Marie Curie, the UK's leading charity caring for people living with any terminal illness and their families. The views expressed are those of the authors and not necessarily of the HIEC, CUHP, NIHR, Marie Curie, NHS or Department of Health and Social Care. The funders had no role in study design, data collection and analysis, decision to publish, or preparation of the manuscript.

**Competing interests:** We have read the journal's policy and the authors of this manuscript have the following competing interests: Both authors were members of the project development team for the Cambridgeshire & Peterborough CCG Project for Data Sharing in End of Life Care (during the stage of the work described in this paper; currently this is the Cambridgeshire and Peterborough ICS Project for Data Sharing in End of Life Care). SB was the project Clinical Lead. The HIEC grant, through which the service development project was originally funded, has contributed to the salaries of both MP and SB in early stages of this research. The data sharing project is a not-for-profit project, currently funded by the Cambridgeshire and Peterborough Integrated Care System. It aims to improve health services for end of life care patients in the locality, as part of the provision of services by the NHS (the UK National Health Service), which is publicly funded and free at the point of care.

implementation; and 3) the parallels it draws with a variety of non-IT, non-health domains of practice as a source of fruitful learning. Teams working on data sharing projects need to prioritise further the shaping of social interactions and structural and contextual parameters in the midst of which their IT tools are implemented. The high number of "ambivalent forces" speaks of the vital importance for data sharing projects of skills in eliciting stakeholders' assumptions; managing conflict; and navigating multiple needs, interests and worldviews.

## Author summary

Sharing data is often effortless in everyday life. As a result, many patients assume that there are well-established and robust systems for sharing their medical records and other data about their health, diseases, and treatments. This is not the case. Patient data sharing systems are notoriously difficult to design, implement, and keep working in the long run. Moreover, while their benefits appear self-evident and significant, the evidence that they follow naturally once data sharing is made technologically possible is uncertain. Here we bring together evidence on the challenges and drivers to patient data sharing from in-depth interviews with 44 health professionals, IT experts, decision makers, patients and carers involved in a data sharing project in end of life care in England, as well as findings from four literature reviews covering 135 studies. We reduce an initial set of 1,800 challenges and drivers to 800+ parameters and, finally, a 22-compoment framework offering new ways of understanding and advancing patient data sharing. Importantly, we find that about three-quarters of the 800+ challenges and drivers are unrelated to the technological solution. Some of the social and psychological barriers to patient data sharing may be much harder to surmount than the technological ones.

## Background

Patient data sharing is a concept of multiple meanings. This multiplicity both contributes to and results from the complexity of endeavours to enable data sharing in practice. Some types of patient data sharing aim to support "individual care", while others seek to "improv[e] health, care and services through research and planning" [1]. The latter drive and rhetoric overlap significantly with those for "open data", "open science" and "open access", yet the patient data being shared tend to be, in contrast, robustly protected. Furthermore, some shared data are used directly by health professionals, while others are first subjected to computational analysis, as in precision medicine [2]. Some shared data are collected in routine clinical practice, while others are generated in research studies.

This paper focuses on *data sharing for individual care where the shared data are collected in routine clinical practice and where they are (to be) used directly by other health professionals and, potentially, patients.* We shorten the full description of the type to "data sharing", as typical in a UK context, and also use the term interchangeably with Health Information Exchange (HIE), as preferred by much of the US and international literature on the topic. We follow a broad definition as per which data sharing for individual care/ HIE "allows doctors, nurses, pharmacists, other health care providers and patients to appropriately access and securely share a patient's vital medical information electronically—improving the speed, quality, safety and cost of patient care" [3].

Multiple benefits of data sharing have been asserted and partially supported by evidence: improved clinical decision making [4,5], patient safety [6–12], coordination and continuity of care[13–15]; reduction in duplicate investigations [4,16–21] and hospital admissions [4,20–26]; holistic management of complex cases [27–29]; enhanced patient experience, involvement, empowerment and activation [12,30,31]; and, importantly, cost-savings [17,20,21,32–35]. Yet despite the indicative evidence, plausibility of the logic models, growing sophistication of the technology, and significant investment in health IT, data sharing solutions face substantial challenges. They often fail to live up to expectations or straightforwardly fail [33,36–47].

This study aimed to identify the challenges and drivers encountered by the flagship type of data sharing project in palliative and end of care in England – Electronic Palliative Care Coordination Systems (EPaCCS) [48] – in the context of challenges and drivers of data sharing more broadly. It aimed to organise them in a comprehensive yet parsimonious framework that underpins specific and non-trivial recommendations for advancing data sharing projects, particularly those in early stages of development and implementation. As the case study project involved primarily adapting pre-existing functionalities of locally used clinical IT systems – selecting and harnessing relevant elements in innovative ways as opposed to designing a solution *de novo* – the paper is more relevant to project managers, policy makers, clinicians and other implementers and early adopters as opposed to IT experts interested in high-level technological innovation.

The study was set in England, which has a National Health Service (NHS) funded primarily by general taxation and free at the point of care [49]. Even if "national", the NHS is far from centralised and integrated. The NHS is an umbrella term for the four health systems of England, Scotland, Wales and Northern Ireland [50]. In primary care, General Practitioners are independent contractors [51]. Hospital trusts, especially Foundation Trusts, have significant autonomy in deciding how demands placed on them are met [52]. Community health services are provided by NHS trusts, community interest companies, social enterprises, local authorities and independent providers [53]. At the time of this study, health care was commissioned and decisions were localised through over 100 Clinical Commissioning Groups (CCGs) [54] (over 200 in 2014 [55]), which have, as of July 2022, been replaced by 42 Integrated Care Systems [56]. Each of these entities has significant autonomy in choosing digital health solutions. This creates a vibrant digital health environment and testing ground for products and systems but also precludes the effective sharing of patient data.

## Materials and methods

This study was part of "Prepared to Share?" [48,57–60] – a mixed methods evaluation of the development, implementation and outcomes of the Cambridgeshire and Peterborough CCG Project for Data Sharing in End of Life Care (the EoLC data sharing project), nested in broader research on patient data sharing and accompanied by a set of impact activities. The EoLC data sharing project, launched in 2014, has been a leading Electronic Palliative Care Coordination System. Details on the broader study, service development project, and the authors' roles in the latter can be found in S1 File.

### Data sources and data collection

**Interviews.**   The interview schedule and approach were informed by principles of episodic [61] and realist interviewing [62,63] and by the guidance on interviews of Brinkmann and Kvale [64]. The interviews sought to identify challenges and drivers to the development, implementation and uptake of the EoLC data sharing project. The core interview schedule and further detail on the methods and theories underpinning the interview approach can be found in S1 File.

Interviews were conducted by MP between October 2013 and October 2015. Data were collected over four periods, reflecting distinct stages of the service development project:

○ initial development, when the project tools and systems were still a work in progress: Oct 2013 – Mar 2014, 10 interviewees, all of whom project team and steering group members.

○ several months after the formal launch of the project (Apr 2014) when GP and specialist services uptake was expected: Aug – Nov 2014, 13 interviewees, 12 of whom GPs and palliative care specialists.

○ stage at which patients and carers were expected to have experienced care involving the sharing of data: Dec 2014 – Apr 2015, 10 interviewees, 9 of whom patients and carers.

○ stage at which the urgent and emergency services were expected to have integrated the data sharing tools into their workflows: Jun 2015 – Oct 2015, 10 interviewees, 7 of whom working in urgent and emergency services.

We did not observe stage-related differences of perceptions of challenges and drivers. Further detail on interviewees whose roles did not fit the stage descriptions (5) or were interviewed outside of the above campaigns (1) can be found in S1 File.

We acknowledge the length of time which has elapsed since data collection (versions of this paper have been in the peer review process of different journals since 2019). We are not reporting on dated issues, such as: challenges around local organisational entities which no longer exist (e.g. Local Commissioning Groups); claims for the need of a "culture change", with the latter achieved by the COVID-pandemic; IT-specific challenges which are likely to have been resolved since data collection due to the level of infiltration of technology in everyday life and COVID-related digital gains (e.g. lack of basic IT skills in some staff groups); and claims associated with events which had prominence in the then social discourse but have now faded away (e.g. the Liverpool Care Pathway scandal in palliative and end of life care).

The sample was purposive, aiming at maximum variation of roles, contexts and types of use of the data sharing tools. Written informed consent was obtained by all participants. All non-patient/non-carer participants received, as compensation for their time, a £30 gift voucher or the opportunity to donate the equivalent to a charity of their choice. The total number of interviews was 40 with 44 primary participants. The interviews lasted between 35 m and 1h 36 m; average length 56 m. The word count of the transcripts was close to 300,000 words. Table 1 and S1 File provide further detail on the sample and recruitment approach.

**Literature reviews.**   Data from four literature reviews on the implementation, uptake (adoption, usage) and effectiveness of health IT were used to query, expand and strengthen the knowledge gained through the analysis of primary data. Two of the reviews – the systematic reviews of Rudin et al. (2014) [33] and Eden et al. (2016) [38] – were on health information exchange (HIE), respectively its usage and effects [33] and barriers and facilitators [38]. The two other reviews – of Rippen et al. (2013) [65] and Cresswell and Sheikh (2013) [66] – were on health information technology. The former was a "targeted review" which offered an "organizational framework for health information technology". The latter was an "interpretive review" on "organizational issues in the implementation and adoption of health information technology innovations". The literature reviews were selected as the most relevant and detailed reviews from the authors' collection of HIE and HIT papers at the time and extensive, but not systematic, literature searches on HIE. The total set from which the four reviews were chosen had over 250 records, either previously saved or kept post-screening of the HIE search results. We did not have the capacity to conduct an accompanying systematic review.

**Table 1. Sample characteristics.**

| Role and number of interviewees | Parameters of variation within sample | Recruitment details and/or unintended features of sample |
|---|---|---|
| **Project developers (6)** | Leadership and managerial roles (2)<br>IT roles with a focus on tool development (2)<br>IT role with a focus on implementation (1)<br>Educational role (1) | MP was member of the service development team and SB was project clinical lead, which provided direct access to the project team interviewees. |
| **GPs – data sharing initiators (7)** (3 further GPs in "out-of-hours and 111" sample and 2 in "Broader project network" sample) | **System**<br>4 dominant system D<br>1 alternative system A1<br>1 alternative system A2<br>1 transitioning from a D to an A2 practice<br><br>**Use of tools for end of life care data sharing**<br>1 – high use<br>3 – use<br>1 – uses a different tool<br>2 – do not use<br><br>**Area**<br>2 – affluent local village<br>2 – remote rural, deprived<br>1 – town centre, affluent<br>1 – inner city, deprived<br>1 – suburban<br><br>**Gender**<br>4 female<br>3 male<br><br>**Career stage**<br>3 early career (1-9 years)<br>2 mid-career (9-19 years)<br>2 experienced (over 20 years) | 45 expressions of interest (through response slip accompanying survey to all local GPs)<br>14 contacted<br>7 agreed and interviewed<br>7 did not respond<br><br>Unintended features of GP subsample (potentially a source of bias but also suggesting "key informants"):<br><br>2 participants with leadership roles in end of life care<br>1 participant with experience of a serious adverse event in end of life care in their practice<br>3 participants conducting own research<br>2 participants with Local Commissioning Group roles<br>1 participant working on another data sharing project |
| **Other healthcare professionals – data sharing recipients (15)** | **Out-of-hours and 111 (telephone triage and advice line)**<br>3 GPs (1 also working in-hours; 2 male, 1 female; 1 mid-career, 2 experienced)<br>3 nurses (all also working in-hours – 2 in general practices and 1 in community; 3 female; 1 mid-career, 2 experienced)<br><br>**Palliative care specialists**<br>3 consultants (2 hospital, 1 hospice; 1 with an IT role)<br>3 nurses (2 hospital, 1 community)<br><br>**Hospital staff**<br>4 of the palliative care specialists (see above)<br>1 Accident & Emergency, consultant<br>1 Medicine for the Elderly, consultant<br><br>**1 GP practice administrator** | Recruitment through:<br>- information in email bulletins of the out-of-hours service<br>- reaching out to participants in awareness raising events and meetings with pilot users<br>- approaching members of the Stakeholder Group. |

*(Continued)*

**Table 1.** (Continued)

| Role and number of interviewees | Parameters of variation within sample | Recruitment details and/or unintended features of sample |
|---|---|---|
| **Patient and carers (9)** | **Patients/ carers**<br>2 patients<br>2 current carers<br>5 bereaved carers<br><br>**Perceptions of data sharing b/n health professionals** (from response slips in recruitment pack)<br>2 – sharing varies<br>2 – sharing good<br>1 – do not share enough<br>1 – question not completed<br>3 – no separate response slip, as joining main interviewee<br><br>**Age**<br>50s – 2<br>60s – 3<br>70s – 3<br>80s – 1<br><br>**Gender**<br>2 male<br>7 female<br><br>**Patient condition** (for interviewed patients)<br>2 – multimorbidity<br><br>**Carer relationship to patient**<br>3 daughter<br>2 wife<br>1 husband<br>1 son-in-law | Recruitment through interviewed GPs.<br>Recruitment materials developed with strong user support. Several participants commented that the experiences described in the vignettes in the recruitment booklet were almost identical to their own.<br><br>8 response slips received.<br>On phoning, 2 patients responded that it was not a good time to participate in research. The remaining 6 were interviewed, along with 3 further family members.<br><br>All patients and carers were from one GP practice yet had varied experiences of data sharing and used a broad variety of services in different parts of the locality.<br><br>Recruitment of patients and carers was discontinued after having interviewed 9 in 5 interviews (original intention was for 24 participants, single person interviews). Data sharing was showing to be a minor and vague consideration relative to their broader experience of illness, healthcare in general and end of life care in particular, with strongly positive opinions expressed in its favour. |
| **Broader project network** (members of the Stakeholder Group and key informants from broader context) **(8)** | **5 commissioners** (project managers or clinical leads in Clinical Commissioning Group or Local Commissioning Groups)<br>**2 Information Governance**/medico-legal issues specialists<br>**1 IT manager** | Recruitment through Stakeholder Group or opportunistically, following meetings or email exchanges aiming to facilitate the service development project. |

The reviews covered 135 source studies. The publication years for the included studies were: 1990-2015 (Eden et al.), 1997-2010 for HIT implementation literature and 1983-2010 for health IT related theories (Rippen et al.), 1997-2010 (Cresswell et al.) and 2003-2014 (Rudin et al.). Both the text of the literature reviews and tables representing source studies were coded line-by-line.

## Interview analysis and synthesis of interview and literature review data

The verbatim transcripts were checked against the audio-recording, corrected, anonymised and annotated by the first author. The analysis of transcripts and their linking to literature review data then progressed through four stages, each of them involving multiple iterations. Data were coded primarily in NVivo (versions 11 and 12) [67], with additional mapping and re-organisation carried out in Microsoft Excel and Word. Coding in NVivo was consistently

"pluralist", following Dupré [68,69]: excerpts were often attributed to three-five parameters rather than one or two most fitting ones. The four key stages of data analysis were as follows:

1. Thematic analysis was undertaken after nine interviews with the six project developers. The aim of that analysis was partly practical: to inform the day-to-day work of the service development team. Both the analysis and feedback on it led to the original analysis framework, which was then expanded through coding the contents of the literature reviews.

2. The resulting framework underpinned the analysis of the remaining 31 interviews, i.e. the analysis moved towards "framework analysis"[70]. Over 1,800 challenges and drivers to data sharing were identified in the process. These were then reduced to a set of over 800.

3. The latter 800+ parameters, together with their best illustrative quotes, were mapped in Excel, with a focus on pairing challenges and drivers. We also developed a framework of six main themes (the IT solution, project team, health professionals, patients, contextual and temporal parameters, and a set of "co-created" parameters), including 29 components across the six themes. Each of the 29 components was accompanied by a numerical coefficient of the ratio of challenges and drivers within it. The coefficient was used to prioritise areas for practical action.

4. As we felt that the above quantitively derived prioritisation was too contingent on our data and the framework still lacked coherence, parsimony and distinctive character, we followed threads from the work on pairing challenges and drivers. We re-coded in NVivo the 800+ challenges and drivers in terms of more formal characteristics, such as whether a challenge or a driver appeared in a pair or stood alone and what the nature of the relationship within a pair was. Finally, we refined the framework and re-attributed the illustrative quotes to it (in Word).

Stages 1, 2 and 3 of the analysis were conducted in 2016-2017. Stage 4 was conducted in 2019, after peer reviews of the original version of this paper.

### Ethics approval

NRES Committee East of England – Cambridge East approved the study (Ref 13/EE/0291).

### Patient and Public Involvement (PPI)

The study was supported by a study-specific PPI group. S1 File reports on user involvement using the GRIPP2 short form [71].

### Results

The framework of challenges and drivers to patient data sharing we propose consists of nine types of "pure challenges", drawing parallels between patient data sharing and other broad and complex domains of sociotechnical or social practice; four types of "pure drivers", defined in terms of whether they were internal or external to the IT solution and project team, and thus the level of control the latter had over them; and nine types of "oppositional or ambivalent forces", with the former representing pairs of challenges and drivers and the latter factors perceived simultaneously as a challenge and a driver. The main text provides a narrative of the key findings, which are illustrated by quotes in Boxes 2, 4 and 6. Extensive further quotes can be found in S2 File. If quotes are minimally attributed in Boxes (e.g. *Interviewee 15*) or unattributed in S2 File, this is to reduce risks of identification. Some roles or combinations of

parameters were either unique for the locality and/or the data sharing project or narrowing down the likely individuals to two-three.

## 1. Pure challenges

Challenges were designated as "pure" if the interviewees discussed them without mentioning an effective counteracting driver. Pure challenges were perceived as immensely complex, able to "kill" a data sharing project, and largely outside the scope of influence of the project team. We formulated nine types of such challenges on the basis of the interview and literature reviews data:

1. radical innovation challenges

2. challenges of working in a health service "on the edge"

3. challenges of working in large impermanent teams

4. challenges of IT work, narrowly construed

5. challenges of reconciling the rules and algorithms of IT with the need for humanity, flexibility and art in healthcare

6. challenges of developing an innovation which enables the core work of some users and peripheral tasks for other users

7. challenges of working with highly sensitive, emotive, potentially incendiary issues

8. challenges at the human-machine interface

9. challenges of reconciling market forces and the public good.

Paradoxically, while experienced as overwhelming in the context of this study, all of the above types of challenges other than 4 (*IT work, narrowly construed*) are shared with many familiar and long-standing domains of sociotechnical or social practice. Historically, a classic example can be the arrival of the railways. Nowadays, it can be the development of a comprehensive recycling infrastructure within a country. Box 1 lists examples of such non-IT, non-healthcare domains for all challenge types. We argue that if the above types of challenges attracted the label of "pure", by virtue of overpowering any effective driver, then patient data sharing projects should consider such external domains as sources of fruitful learning.

### Box 1. Examples of areas and endeavours outside of health IT which experience one or more of the pure challenges

#### Radical innovation challenges

Key features:

- system-wide innovation which needs to filter through to every unit within the system;

- significant behaviour change involved;

- life-changing impact as a result of changes to material structures and behaviours.

Parallels outside of health IT:

- development of large-scale recycling infrastructure;

- water and sanitation systems in the developing world;

- (historical) the arrival of the railways, the construction of the modern road infrastructure, the growth of the aviation industry, etc.

## Challenges of working in a health service "on the edge"

Key features:

- the challenges experienced by the healthcare system are replicated in the projects that involve it;

- the challenges experienced by the healthcare system limit the capacity of organisations, teams and individual health professionals to engage in innovation, even if that innovation aims to resolve precisely those challenges.

Parallels outside of health IT: possibly local governments and national bureaucracies.

## Work in large impermanent teams

Key features:

- an ambitious common goal leads to the creation of new teams and/or organisations;

- these are held together by a loose hierarchy and limited shared lines of reporting and accountability;

- the primary belonging of team members is outside of the team/ organisation that works towards the common goal.

Parallels outside of health IT: supranational, multilateral organisations, such as WHO, UN or the European Union.

## Rules, laws and algorithms in spaces also calling for humanity, flexibility and art

Key features:

Contrast between:

- the "orderly nature" of IT – of rules, algorithms, logic, rationality, 1s and 0s, clear boundaries, fixed choices that open pre-determined pathways, etc.

- the "messy nature" of healthcare – characterised by complexity, uncertainty, suffering, embodiment, humanity, intuition, creativity, and "the art of medicine" in addition to medical science.

Parallels outside of health IT: the application of laws, governance rules, bureaucratic processes, etc. in messier or more creative areas of life, such as the arts, interpersonal relationships or humanitarian work.

## Core tasks – peripheral tasks

Key features:

- the intended users are both experts/specialists and non-experts (generalists, other types of specialists, as well as "low-skilled" workers);

- specialists are expected to lead the way and provide support, but are too few;

- non-specialists are numerous and of various professional backgrounds, various hierarchical levels, working in various services, with various levels of pre-existing knowledge, learning needs, ability to shape their roles by themselves, and motivation to become engaged in professional development that is not crucial to their role;

- in data sharing projects, the divide pertains both to IT skills and domain skills (e.g. end of life care).

Parallels outside of health IT: any field where top specialists and non-specialists, experts and lay users, novices and advanced learners of various levels need to interact. Any field where individuals for whom an issue is of primary concern and a skill is a core skill are collaborating with users for whom an issue is of secondary concern and a skill is a peripheral or optional one.

## Work with highly sensitive, emotive, incendiary issues

Key features:

- minor missteps can grow into an unmanageable scandal that "kills" a project;

- a rational, balanced response may not be possible and may need to give way to a strategy of appeasing fears and softening resistance;

- parties who can benefit from the scandal will actively seek to do so;

- most of the parties facing reputational damage will seek to distance themselves from a project, even if they have been close collaborators.

Parallels outside of health IT: any issue that triggers conflicting views about right and wrong, good and bad, natural and unnatural (e.g. sexuality, abortions, immigration, euthanasia, etc.); any issue that touches on deep fears and/or fundamental values (the good life, death, money, identity, home, personal property, human rights, etc.).

## Technology – human users

Key features:

- technology can only fulfil its potential if handled by competent, well trained and responsible users;

- technology needs to be adapted to what is most natural and intuitive for users as most will not go to great lengths to learn how to operate it well;

- technology can be of little use, fail or even become dangerous if not operated well.

Parallels outside of health IT: engineering design; ergonomics; health and safety in the operation of dangerous equipment.

## Market forces and public good

Key features:

Tensions and overlaps between:

- the presence of powerful market forces in healthcare is problematic as motivations towards improving people's health and well-being can come into conflict with profit-making motives;

- the presence of market forces in healthcare can improve efficiency, value for money and accountability in public services;

- voluntary work, social enterprises and university-led research projects can be positive balancing influences in public services when the latter are expected to be more and more responsive to market forces;

- practically any voluntary, pro bono or "free" work brings in demands and underlying agendas of its own.

Parallels outside of health IT: Any field where strong profit motives can skew the field's primary goals and means (education, arts and culture, religion, etc.) although there is no in-principle contradiction between making a profit and the values of that field. Alternatively, any field where work that is normally highly paid is done for free.

Below we summarise subtypes and points raised in relation to all nine types of pure challenges, with a focus on the first four, which were discussed in most detail by the interviewees. In the narrative, we alternate between using a past tense, when discussing more directly the study findings, and a present tense, when we represent the conceptual framework in a more abstract way (note though that every element of the framework was abstracted from study data, even if further interpreted).

## 1.1. Radical innovation challenges

Of all pure challenges, the challenges of radical innovation were represented through the largest number of distinct subtypes: 1) magnitude of repetition involved in implementation; 2) multiplicity of available alternatives; 3) vicious circles likely to be experienced during innovation implementation; 4) uneven involvement of different stakeholders; 5) overvaluing the uniqueness of one's work; 6) securing external support (e.g. a sponsor or host); 7) impact of delayed external innovation; and 8) slipping timelines. Below we discuss in some detail the first four of the subtypes, as these received the most attention in the interviews.

**1.1.1. Magnitude of repetition.** A key reason why radical innovation projects fail, become stymied, or advance arduously slowly, even if the solution they are offering is compelling, is the immense magnitude of repetition involved in implementing them. Numerous repetitions are needed both in setting up and maintaining the underpinning infrastructure (in multiple settings, teams and computers) and in modifying the behaviours and mindsets of stakeholders. Examples of such work in the EoLC data sharing project included, amongst others: persuading organisations to take part; negotiating the parameters of their involvement in the context of omnipresent conflicting demands and opportunities; the signing of Information Governance (IG) agreements with each participating organisation; configuring the systems of each organisation and, sometimes, every computer within the organisation; developing setting-specific launch activities and communications; the training of staff; tailoring aspects of the IT solution to the needs of each organisation; and negotiating continued involvement in cases when such tailoring was problematic.

The data sharing project team expected that much of this work would be owned by project users, many of whom, in contrast, expected that it would be done or at least managed by the data

sharing project team. Not infrequently, tasks intended to be user-owned were taken up by members of the project team outside of their lines of duty, as this was the only way to advance the project.

The magnitude of repetition also means that innovation has an unavoidable "dark side". Key aspects of it are mundane and repetitive. Yet most individuals working in innovation are drawn to its excitement, creativity and buzz and may struggle with its mundane aspects.

**1.1.2. Alternatives.** Between themselves and unprompted by a specific question in the interview schedule, the interviewees identified 13 *types* of alternatives to the EoLC data sharing tools, mostly EoLC-specific tools (e.g. patient-held "yellow folders" containing EoLC information) or broad-based tools which have modules with EoLC information (e.g. the national Summary Care Record). Most of these alternatives had subtypes or periodically updated versions. Most were standard practice in some organisations and workflows. The interviewees did not single out any such alternative as consistently better than the EoLC data sharing project, but most were spoken of as having some major advantages; as embedded in current workflows; or as a promising new opportunity to explore. Together, however, they led to confusion, frustration and refusal to engage in certain types of data sharing, end of life care work, or work at the intersection of the two.

**1.1.3. Vicious circles.** To be successful, radical innovation also needs to break away from a range of vicious circles, or at least resolve persistent tensions before they become vicious circles. The interviews suggested five main types.

First, users are prone to give up quickly due to disappointment with a tool, while developers need initial grace, continued engagement, and constructive feedback. Second, the emergent nature of innovation comes into frequent friction with traditional project management requirements for well-specified and accountably adhered to objectives, roles, responsibilities, risks and timelines, amongst other. Third, and crucially in the context of healthcare, the time and cost savings and the streamlining of work for overburdened services, as promised by data sharing, first demand significant investment as well as reorganisation of work processes. Yet overwhelmed clinicians and organisations may find it close to impossible to invest in training and reorganisation vis-à-vis the needs of suffering patients and current organisational targets.

Fourth, an innovation may offer dramatic improvement relative to current processes and structures, yet if the latter are "not broken", they remain "good enough" for many users for a long time after a new and much improved solution has become available. The old persists through the pull of habit, familiarity, and integration with other processes and structures.

Last but not least, success can be self-defeating. Recent or current successes in a field can result in disinvestment of funding and attention, as opposed to a drive to improve further through innovation. For instance, the area of the EoLC data sharing project has long been a national lead in end of life care. This made end of life care a prime candidate for deprioritisation in the context of severe resources limitations.

**1.1.4. Uneven involvement.** Even the most inclusive stakeholder group on a patient data sharing project cannot include representatives from all relevant settings and teams: over 330 was an inclusive, but not exhaustive, count of these for the EoLC data sharing project [61]. A project team is deeply involved with some settings and teams. It has on and off communication with others. The involvement of still others is intentionally delayed, as part of a staggered implementation approach. Finally, some stakeholders are unwittingly excluded. Representation does not equal effective representation either. A representative on a stakeholder group may not be able to cascade information down successfully (e.g. if they do not have sufficient power within their organisation) or may not escalate it up from the lower levels (e.g. if they do not do sufficient frontline work or the staff-manager relationship is strained).

The remaining four subtypes of radical innovation challenges (overvaluing the uniqueness of one's work; securing external support; impact of delayed external innovation; and slipping timelines) are presented briefly in S3 File. Relevant quotes can be found in S2 File.

## 1.2. A health service "on the edge"

The challenges inherent in the very context of (UK) healthcare appeared to be the most prominent topic in the whole study, even if not data sharing specific and even if emerging without a specific prompt in the interview schedule. These fell into four main types: 1) resource limitations, 2) fragmentation, 3) constant transformation, and 4) environment deprioritising "extras"/ only focusing on "core business". All those challenges are not only well known in health services research but are often elements of the everyday social discourse on healthcare. Here we describe only issues of fragmentation, as they were most often explicitly linked to the particularities of the EoLC data sharing project. The remaining subtypes of challenges are presented briefly in S3 File. Relevant quotes can be found in S2 File.

The high level of fragmentation within an otherwise national system was identified as a source of generic but, potentially, uniquely harmful challenges, as data sharing projects are by definition cross-organisational. Mostly, the fragmentation arises from and feeds back into the significant levels of autonomy of organisations and, to some extent, professions within the UK health service; the lack of clear mechanisms for driving multi-setting projects; the multiplicity of simultaneously running and often overlapping projects; and the lack of sufficient central/ national leadership.

A range of further aspects of fragmentation were discussed less frequently. The targets of different organisations could come into conflict with one another. Health professionals often lacked an overview of a patient's end of life care journey and/or understanding of the role of their colleagues in it. The requirement for involvement in many cross-organisational projects was perceived as problematic: involvement may not be resourced from new funds but still be expected on the grounds of "we are the NHS". Not only patient services were reported to be fragmented, but also local structures aiming to provide organisational support, such as IT support teams or training initiatives in end of life care.

The impact on data sharing of the health service fragmentation was experienced even more acutely in geographical and administrative boundary areas, which were often moving backwards and forwards in their belonging to local healthcare structures. Also, services with a broad geographical coverage, such as the major referral hospital for the region, had large numbers of patients not covered by the EoLC data sharing project, as such patients were coming from "other" Clinical Commissioning Groups.

## 1.3. Work in large impermanent teams

The third most frequently mentioned type of pure challenge arose from working in a large impermanent team, which had a loose hierarchy and limited shared lines of reporting and accountability. Participants in multi-setting data sharing projects have a key common goal. Yet they also have further sub-goals for data sharing that would benefit their organisations and staff while variably benefitting, making no difference to, or endangering the goals and practices of other organisations and teams. In this study, collaboration between organisations and teams within the health service was seen overwhelmingly negatively. (Note though that this negative perception did not apply to the core project team, see "2.2. Pure drivers internal to the project development and implementation team".)

With minor exceptions, the discussion of this subtype of challenge remained at a highly specific level. Interviewees expressed frustration with specific people or organisations, without considering the possibility that these were presenting as "difficult" because of deep-seated, structural problems of interaction amongst elements of the health system. Issues discussed most often included: "politics"; the importance of watching out for the interests of one's own team and organisation; the lack of action to match promises given; and differences in language. The language of IT was, expectedly, perceived as a particular challenge.

Most stakeholders involved in the data sharing project belonged to external teams and organisations (relative to the core project team). Their contribution was unpaid for by project resources. This created precarious involvement arrangements. There was always a background risk that an individual or team might disengage. Concerns about disengagement also limited the project leads' capacity to manage or arbitrate in cases of conflict or performance falling below expectations.

Disengagement could be particularly hard to detect from individuals and organisations that were inconsistently represented on the working group. In several interviews it became clear that actions agreed upon or even proposed by some such organisations were not initiated. At the same time, the project team were progressing the work as if such actions had been completed or were at least underway (see "Uneven involvement" above).

### 1.4. IT work, narrowly construed

The multiplicity of non-interoperable clinical IT systems and the complexity of Information Governance (IG) were the two leading challenges in the *IT work, narrowly construed* category, in terms of degree of attention given to them by interviewees. IG is not discussed here, as the level of detail around it demanded a separate paper.

Arguments around choice, variety, and competition as drivers of continuous improvement are taken for granted in many contexts. Yet if a health professional interviewee commented on the variety of clinical IT systems, it was in terms of it being at best incomprehensible ("it's just daft, isn't it"; "makes no sense" *[Out of hours professional 4, GP]*; "I just don't know how that could have happened, but I'm sure there must be some vested interest" *[GP 7]*; "it's a shame"; "just seems bizarre" *[Out of hours professional 1, Nurse]*).

A small number of interviewees warned that technically enabled integration may not be meaningful practical integration. For instance, the movement from one "integrated" system to another could be unidirectional. Alternatively, the forwards-backwards switching was not necessarily possible without complex, and potentially unsafe, workarounds.

In addition to the multiplicity of systems and their lack of interoperability, several further and familiar aspects of a digital environment lacking maturity were discussed. These included: unresolved challenges around mobile working; "stickiness" of old clinical IT systems and software more broadly; the lagging behind with the implementation of comprehensive IT solutions in key settings; the presence of pockets across the health service where paper notes were still the norm; or generic signal issues.

Inadequate users' knowledge and skills were also identified as a challenge. These related mostly to generic IT skills, particularly in some staff groups, and a limited understanding of record sharing. Importantly, it was impossible to convene a project team where all members would have high levels of IT skills in addition to their primary expertise, whether in a clinical area or project management in the health service.

Rarely, it was pointed out that IT systems are, after all, "work in progress". Even the best amongst them may not necessarily live up to their promise; may not be improved upon fast enough; and ultimately fail for the needs for which they were sought, developed or bought, often after significant financial investment.

### 1.5. Rules, laws and algorithms in spaces also calling for humanity, flexibility and art

In discussing data sharing and health IT more broadly, several interviewees touched on their "orderly nature" – of rules, algorithms, logic, rationality, 1s and 0s, clear boundaries, etc. – and the ways in which it did not match the nature of health care. The latter was perceived as

anything but orderly. The organisation, pathways and decision-making processes of healthcare are exceptionally complex. At times, they are not only complex but potentially irrational, primarily because of the endurance of historical patterns of organisation of services. Crucially, healthcare deals with embodied humans whose pain, suffering, needs and behaviours cannot be easily slotted into templates. It often requires as much art and creativity as it requires science and logic.

Even health professionals who were self-professed "techie" and loved "fiddling" with computers and new software, expressed significant frustration with some of the set-up around data sharing, with the sequencing of operations in the software being poorly synchronised with the flow of a clinical encounter. Some were proactive to engage with developers in looking for resolutions to challenges, but a positive outcome was not a given.

At other times it was less clear whether the clinical IT system was not "smart" enough in mimicking real-life processes or whether those processes were unnecessarily complex, with the challenges of representing them in an IT tool serving to highlight an improvement need.

### 1.6. Core tasks—Peripheral tasks

The EoLC data sharing project was expected to serve both experts/specialists in palliative and end of life care and non-experts, such as General Practitioners, other types of specialists, and even the so-called "low-skilled" workers, such as care home staff. It was an example of a project where the specialists are expected to lead the way, but are too few, and where the non-specialists are numerous, with various levels of pre-existing knowledge, learning needs, ability to shape their role by themselves, and motivation to become engaged in professional development that may not be crucial to their role. The challenge of simultaneously meeting the needs of users for whom an area of work is a core task or a peripheral one came from two directions – both end of life care and data sharing.

### 1.7. Work with highly sensitive issues

In a similar way, the project was a double exemplar of work which carries the risks of touching on highly sensitive, emotive, incendiary issues: death and dying on the one hand and personal space, privacy and confidentiality on the other. Respectively, the knowledge and skills needed for a successful, thorough implementation were complex and time-consuming to learn.

### 1.8. Technology—Human users

A little discussed but well-defined source of challenges concerned the interaction between technology and users, with the former, no matter how automated, doing nothing fully by itself. The challenge was particularly prominent with regard to the quality of clinical record keeping and especially its underpinning coding.

### 1.9. Market forces in public services

Challenges around the presence of market forces in a publicly funded health service were not specific to the EoLC data sharing project (which was non-commercial, funded by a research and innovation grant) but had an impact on it too. Perceptions of the impact of market principles and forces were contrasting: some interviewees saw them as infiltrating healthcare in ways that were harmful to its goals, while others saw them as insufficient for what it needs. In the whole study, no other name of an organisation or person external to the project was mentioned as frequently as the name of a UK supermarket chain – both as an example of efficiency in collecting and using consumer data and as a face for the fear of data sharing.

Box 2 and Box 3 offer, respectively, illustrative quotes and recommendations associated with the challenges discussed above. S2 and S3 Files provide further detail.

---

### Box 2. Illustrative quotes about the nine types of pure challenges

#### 1.1. Radical innovation challenges

Examples of the subcategories of "vicious circles" and "magnitude of repetition"

[O]bviously, the information has to be on it for, I don't know . . . 90% of the patients? So you're almost sure to get something useful out of it if you look in . . . [I]f we got into the patient record and it consistently wasn't there, we do say and think, 'well, what's the point?!'. So the information has to be on there and the access has to be easy. *[Interviewee 15, palliative care professional]*

[I]t's a bit demoralising when you find the ambulance service won't look at it [template]! . . . [Y]ou just think we've got better things to do, because if you have a rat and you want to motivate a rat to fill that in, you'd be giving the rat some praise . . .! *[Interviewee 39, out of hours GP]*

So we get eight different MDT [multidisciplinary team coordinator] projects instead of an MDT project with perhaps a little bit of local variation. It very much feels that you get eight different ones. *[Interviewee 6]*

#### 1.2. Health service on the edge

Example of the subcategory of "staff shortages"

[T]hey're absolutely cash-strapped . . . They are a very, very challenged organisation and asking more from them in a field that the CCG sees as top priority and still piles pressure on them to do other things, we are dealing with what is regarded as a failing organisation as our main platform of making this work. *[Interviewee 11]*

I get really upset talking about this [pause]. [H]e'd fallen out of bed and she couldn't get him back to bed, so when I came back . . . that day on the Sunday afternoon, I went round to see him and he was dead and he was on the floor, because she couldn't lift him and because the district nurses had said again that he wasn't important enough.

[T]he reason for this was not because they are awful people who hate my patients. It's because the community nursing service are based elsewhere and it takes half an hour to drive out here and see somebody, and then half an hour to drive back, whereas if they're in [cities], it takes them 10 minutes to drive somewhere. *[Interviewee 13, GP]*

#### 1.3. Work in large impermanent teams

Example of the subcategory of "politics and conflict"

*Interviewee*: It has not been a project is maybe what I will say.

*Interviewer*: . . . And it has been what?

*Interviewee*: A game. . . . Where tactics seem to be required. Of course it has been a project, I can't really deny that, but finding ways of coordinating the diverse interests and using resources sensibly . . . has defeated me. *[Interviewee 11]*

### 1.4. IT work, narrowly construed

Example of the challenge of convening a project team where all members have high levels of IT skills in addition to their primary expertise

I very specifically put into my email, 'I am there to do the underpinning knowledge, I am not IT-minded at all, so you can ask me a few questions about the IT and I'll be able to answer them from what I've picked up, but I am not an IT brain'. *[Interviewee 4]*

I know from meetings where [names] have gone to . . . demonstrate the template, and, unfortunately, it was understood to them that it was a magical system . . . and [GP practice], who are very competent [System D] users and have been for nearly eight years, just laughed, laughed them off and just said 'this is crap, we're not using it'. *[Interviewee 3]*

### 1.5. Rules, laws and algorithms in spaces also calling for humanity, flexibility and art

Example of the subcategories of "mismatch between the IT representation and the clinical or practical reality" and "use of unsafe workarounds"

Our call handlers have to tick buttons and say things like, 'no clinician available', when in fact there is one, 'put into queue', when in fact they're not going to be put into queue. Those calls are only going to be warm transferred, and so they're having to click buttons that say the exact opposite of what they're doing. It's a way of forcing the software to do something that it's not designed to do, and we can do it, it works. *[Interviewee 36]*

### 1.6. Core tasks – peripheral tasks

[H]opefully, if our GP or district nurse colleagues have had that experience of a surprise answer [of a patient's preferences for end of life care contradicting their expectations], then they've realised how incredibly helpful it is. But I know quite a lot of them do like the specialist nurses or people from the hospice to have those discussions. *[Interviewee 35, Palliative care professional]*

Because I'm not in mainstream general practice now, I don't know what the understanding of it [data sharing] is out there . . . [P]art of the problem was poor GPs have got so much to get their heads around that they're probably not bothered, and if they're sitting in their own little world, it doesn't bother them too much. *[Interviewee 36]*

### 1.7. Work with highly sensitive, emotive, incendiary issues

You have the Daily Mail, don't you, '[Supermarket chain]'s staff will be able to see your data!' and then people, 'Ooh, I'm not doing that!'. And there is a massive lack of understanding out there in the general public of what it's actually for. *[Interviewee 37, out of hours nurse]*

Maybe it's bigger than this, maybe it's bigger than the local. Maybe it's the demoralisation of the collapse of the Liverpool Care Pathway, and many of us that thought that it was a good tool . . . And the adverse media attention and the misuse of it perhaps in the district general hospitals, by untrained staff, and poor communication, it's rather given it a bad name . . . [A]dvance care planning has got harder since that publicity. *[Interviewee 14, GP]*

### 1.8. Technology – human users

[Y]ou can try and say, 'I only want . . . clinical events which are a face-to-face thing from a clinical member', but quite often doctors will not change the thing which says where it occurred . . . [I]t's not a smart system, it's just doing what you tell it, to check what the last three things that get dragged in are. *[Interviewee 21, GP 7]*

[L]oads of people are generating data but we don't have the discipline, and we don't have the systems, and we don't have the culture which sees it as necessary to be strict enough at the point of entry to generate stuff which makes a difference further down the track. *[Interviewee 9]*

### 1.9. Market forces in public services

But that's because the NHS is so desperate, nationally, that they'd rather have a consultancy company coming in for three months, doing its bish, bash, bosh, 'this is how we're going to cut three million, five million, 10 million, 30 million off your budget and you all have to do that'. And then three months later, they've left, they're gone and we have to pick up the pieces and then start again! *[Interviewee 16]*

I struggle with the thought that [supermarket chain] can tell me how often I buy bananas, yet we can't work out a way of holding information to make somebody's End of Life Care better. *[Interviewee 4]*

---

### Box 3. Shortlist of recommendations for action based on findings about "pure challenges"

#### Map meticulously – organisations, settings, contact persons, alternative tools and processes, IT systems . . .

Map as comprehensively as possible the **entities** that need to be reached (organisations, teams, settings); the **alternatives** to the solution you are developing and implementing; and the variety of *local* **IT systems**.

Do not underestimate the scale of the task. An inclusive but non-exhaustive count of the settings and teams that needed the tools of the EoLC data sharing project, in a single region, was 330. Those tools were also found to have a minimum of 13 *type*s of alternatives. None of those are static either.

#### You'll often progress one step at a time

Be aware that in many localities (at least in England), many healthcare organisations will need to be engaged in a data sharing project **one by one** rather than through a reliable national or local mechanism for implementing health IT projects.

#### Don't lose the big picture in the step-by-step work

Even if involvement needs to be staggered, do not lose sight of the organisations left "for later stages". Months, easily turning into years, of work excluding such organisations can result in processes and structures that are no longer suitable for their involvement. A data sharing project is as strong as its weakest link.

---

### Introducing innovation involves a lot of mundane repetition which happens at all levels of authority and skill

Take into account that some of the repetitive work on introducing an innovation requires highly skilled communication and negotiation at top hierarchical levels (e.g. for securing buy-in); some is low-level technical (e.g. setting up computer systems); and some is in between (e.g. refining the IT tools).

### Have a plan for the vicious circles

Involve initial users as co-developers, not as consumers who must be quickly satisfied. "Initial users" may be users over the first few years.

There is a brutal paradox that, in most cases, the better a data sharing tool performs, the greater unmet need and opportunities for further improvement it will reveal.

### Appealing to ideals and visions for the future is not a substitute for appropriate resourcing

While data sharing projects argue they can resolve some (many) challenges, they will first exacerbate some (many) challenges, such as the need for more staff time and developing new ways of working. Under-resourcing such projects commits them to failure. "Radically improving care" for patients is not a sufficient argument. The power of ideals is in short supply, all the more in a post-pandemic environment.

### Enable the easy reporting of mismatch between clinical and virtual reality

Create accessible and reliable routes for user feedback. Prioritise work on removing the need for workarounds with safety implications or where A is represented as non-A (so that "the computer" lets one move on).

### Attract boundary spanners

Plan for working with a broad diversity of personalities. Create formats of interaction that improve understanding of each other's roles and working styles and reduce stereotyping. Involve, as much as possible, staff with go-between roles and excellent boundary-spanning skills.

### Expect scandal

Data sharing + healthcare is a combustible combination. Expect misunderstanding and scandals. Learn from work on managing scandal and responding to rumours.

### Commit to scaling up data that has nuance and is of high quality

Solicit and present data on the entities engaged in a data sharing project as a proportion of all entities that need to be engaged, not only as raw numbers. The latter can be hugely impressive, and the work done needs to be celebrated, but the former (proportions) are the meaningful indicator of coverage. Similarly, solicit/ present data on levels of engagement. Having access to the tools is different to using them. Using them once a month is different to using them daily.

## 2. Pure drivers

We call "pure drivers" those drivers which were not invalidated by claims of opposing challenges or this was occurring only minimally. The 4-component framework of pure drivers we propose is defined in terms of whether the driver is internal or external to the data sharing solution and the team developing and implementing it. The degree of internality/ externality is associated with a greater or lesser control of the project team over a driver as well as greater or lesser importance of the quality of the IT solution for its broad uptake. "Internal" in relation to the IT solution was conceptualised as pertaining not only to its functionalities, user interface and technical specifications, but also to its clinical domain and goals (what you can do and achieve with it). "Internal" in relation to the project team concerned its leadership (a project manager and a clinical lead) and core team members (further five of them).

### 2.1. Pure drivers internal to the IT solution

Of the types of factors which propelled end of life care data sharing forward and which were internal to the IT solution itself, the interviewees highlighted: 1) the vision for the solution and its expected benefits; 2) the quality of the solution in terms of functionalities, performance and user friendliness and 3) the nature of the clinical domain.

Of these, the vision for and expected benefits of data sharing were by far most frequently addressed. Amongst themselves, the interviewees identified almost 60 major and minor advantages of the broad sharing of patient data. Major perceived advantages were:

○ avoidance of unnecessary and unwanted hospital admissions

○ improved medication decisions and increased confidence in making them

○ reduction of errors and improved patient safety

○ benefits to coordination and communication in broad multidisciplinary teams

○ difficult conversations at the end of life handled more sensitively, without shock or repetition

○ duplication of work reduced

○ resources used more efficiently and processes made more reliable and smoother.

The brevity of the core data sharing template (where users who initiated a sharing process could enter data, some of them prepopulated) was repeatedly highlighted, alongside its simplicity, intuitiveness and good structure. They obscured all other functionalities, features and performance parameters of the broader solution.

Importantly, the nature of the thematic domain – palliative and end of life care – was seen as facilitating high level of commitment to data sharing.

### 2.2. Pure drivers internal to the project development and implementation team

The personality and leadership style of the project clinical lead, the composition of the core team, and the working style of its members as individuals and as a group were, with minor grievances, evaluated highly positively by both team members and external collaborators.

This positive overall picture did not mean that conflict and tensions were not present. The latter were more notable in early stages of the project and also appeared disproportionately more often in expanded formats of the team when, for periods of time, external colleagues were involved intensely to advance the project in specific ways or settings. Such challenges

were partly discussed under "Large impermanent teams" above and are considered again under "Ambivalent forces" below.

### 2.3. Pure drivers at the interface between the internal and external

Significant work was required by team members to make the project fit with a range of features and requirements of the external environment, including: a national information standard for data sharing in end of life care; national and regional service monitoring and reporting requirements; local work on standardising data sharing; the local information management strategy; and local incentive structures (e.g. through adding fields to the template that would make its use the easiest way to implement a positive process and also report on it). Such work gave credibility to the project. It also aided its pragmatic attractiveness by allowing the simultaneous achievement of numerous goals if its tools were taken up.

Apart from the IT solution, the team or individual members also needed to "fit with" key structures in the external environment. Team members brought in pre-existing networks and connections or the "go-between", "boundary-spanning" nature of their role as resources to the project. They also did new work to link with and influence external structures capable of advancing or slowing down the project.

### 2.4. Pure drivers external to the IT solution and team, but internal to the broader health IT ecosystem

In a small number of cases, interviewees mentioned the beneficial impact of developments within the broader health IT ecosystem, happening without any involvement from the core project team. These included the development of new functionalities in the main clinical IT systems; the trailblazing work of other data sharing projects; and the thoughtful practices of certain organisations and health professionals around consenting patients to data sharing. Most importantly, such a pure driver was the "crest of the wave" of interest in data sharing across the health economy, which had grown exponentially since the beginning of the project. This included some highly positive attitudes of health professionals and patients.

Box 4 and Box 5 offer, respectively, illustrative quotes and recommendations associated with the pure drivers. S2 and S3 File provide further detail.

---

Box 4. Illustrative quotes about the four types of pure drivers

2.1. Pure drivers internal to the IT solution

Examples of the subcategory of "ideal vision for the solution and its outcomes"

### Key expected benefits
### Unnecessary hospital admissions, especially when unwanted by patients, avoided

[I]t's rare to find a frail older person, an 85-year-old, dying or otherwise, relishing the idea of a hospital admission, quite the opposite . . . [T]hey fall and then they don't get given a choice because they can't get up, so the ambulance crew just carts them off, but they'd much rather stay at home. And unless we get the right information available at the right time to the ambulance crews and to the Emergency Departments, we're not going to reduce these unnecessary trips to hospital. *[Interviewee 31]*

---

### Improved medication decisions and increased confidence in making them

Prescribing Diamorphine is quite scary stuff, and if you can see that somebody is already on a syringe driver and you're only being asked to re-prescribe what they're already on, then that feels way more comfortable than [when] you get a phone call out of the blue about somebody you know nothing about. *[Interviewee 40, out of hours GP]*

### Errors reduced and patient safety enhanced

I saw a chap last night, I knew he was unwell, had an abdominal mass, I needed to see his . . . scan. . . . [L]uckily I go over and ask somebody else to show me, but the systems still not talking to each other is the biggest hindrance for patient safety. *[Interviewee 37, out of hours nurse]*

### Benefits to coordination and communication in broad multidisciplinary teams

I've been around to see another patient who's palliative and . . . he said somebody had come round the day before . . . and he didn't know who this person was, and I didn't know who this person was. I think it was a Macmillan nurse . . . [I]f it was all documented, I'd be able to see that it was so-and-so who'd gone in the day before and made this plan and they'd have also been able to see that I had rung social care because he wasn't managing his medication. *[Interviewee 17, GP, alternative system]*

### Difficult conversations with patients handled more sensitively, without shock or repetition

Then we had to go back to [hospital] again after he had the tests . . . and we spoke to a young person there, and she was very nice, and she said, 'Of course you realise you've got cancer', just like that. 'Yes, okay' and we looked at one another. *[Interviewee 28, recent carer, wife of deceased patient]*

### Duplication of work reduced

DNR forms we could have . . . the local nursing home one [as well as] the red-rimmed standardised [area] one, but we would then have to email every one over to the Ambulance Service. We would also then have to code it on [System D] and then share that as patient special notes with the out-of-hours service. . . . I would rather [we have] a single record and we let everyone into that with consent when agreed. *[Interviewee 21, GP, dominant system]*

### Resources used more efficiently and processes made more reliable and smoother

It saves a lot of time and frustration and all the rest of it so as long as people read what's there, then it's hugely, hugely helpful. Yeah, you can see I'm a fan of it! *[Interviewee 35, Palliative care specialist]*

[T]hey'd be spending, what's that [sigh], A&E attendance, what, £250? If it's more complex, if he stayed more than 4 hours, he's going to be £2,500 bill for the CCG . . . And that £250, I might just as well go home and burn it in my fire at home for the benefit that it's doing anyone. *[Interviewee 39, out of hours GP]*

### 2.2. Pure drivers internal to the project development and implementation team

Examples of the subcategories of "lead and leadership style" and "working style of team members and team as a whole"

[O]ne of the greatest strengths is obviously [name] and his leadership of the project. He has a very calming and considered way of working, he's always very validating of everybody's contribution to the project and in that way he's really got a good team around him who are all really positively motivated towards delivering this. . . . [He] takes the thorny issues and he deals with them head-on so they don't become barriers, but he does it in such an affirming way with all of the people that he works with that he is managing to carry this project forward. *[Interviewee 1]*

I just feel it's very much a group of, 'We work for a solution, let's find a solution, there's a problem, there is [a solution]' and that's very refreshing, I must say, I think it's been very good [laughs]. . . . [I]t would probably look different had it been a CCG-run project, because . . . the funding would have looked different . . . but also the [human] resources . . . there wouldn't have been the same level of people. *[Interviewee 16]*

### 2.3. Pure drivers at the interface between the internal and external

Examples of the subcategory of "fit with and work on adapting features of the IT solution to a variety of dimensions of the external environment"

[We] try to join things together with [location], because most of the elements that were trying to be captured were the same and trying to, which is our general aim across the CCG, is to standardise the way things are recorded, and End of Life was another one like that. . . . [W]e saw Share My Care as a part of an ongoing programme of work around End of Life recording, but, of course, it introduced the concept of sending the summaries to the out-of-hours. *[Interviewee 6]*

Some of this will just be trying to key ourselves into the information management strategy of the CCG . . . but understanding, I guess, what we're delivering over what timescales and then marrying that up with the information management strategy and . . . any other initiatives that we have. *[Interviewee 2]*

### 2.4. Pure drivers external to the IT solution and team, but internal to the broader health IT ecosystem

Example of the "crest of the wave" subcategory

Just last week I had one of the consultant oncologists here . . . and he somewhat timidly said to me, 'you know, I gather that lots of you GPs out there are using [System D], do you think there might be any way and would the CCG possibly approve if we were to arrange to get access to [System D]?'. And when I bit his hand off and said, 'I'm desperate for you to get access to [System D]!', he couldn't quite believe that we weren't going to be difficult but for my response to be, 'this is exactly the way we want things to go'. *[Interviewee 10]*

### Box 5. Shortlist of recommendations for action based on findings about "pure drivers"

#### Build more solid reality-vision bridges

Articulating and sharing inspiring visions for the effective sharing of patient data is crucial. Yet it is also crucial to draw bridges from current realities which may be depressingly far from the vision.

#### Evaluate rather than "evidence". Do not make the evidence the overriding arbiter of a project's future

Rigorous evaluation projects need to be part of the work on developing data sharing solutions. The power of evidence in driving forward the implementation of innovation is well known, resulting in attempts to "evidence something" rather than conduct rigorous research and evaluations of it.

That said, negative evidence form early stages of the work should not, by itself, lead to discontinuation of promising projects. Positive outcomes of innovation in complex contexts are difficult to capture. They may take years to surface and accumulate, not least because the "final" version of the innovation is still emerging.

#### Be strategic about "fitting in", not only about "standing out"

Data sharing projects need to plan for a number of sub-projects ensuring a "good fit". For instance, in the EoLC project, such work addressed:

○ a national information standard for data sharing in End of Life Care

○ national and regional service monitoring and reporting requirements

○ local work on standardising data sharing

○ the local information management strategy

○ local incentive structures.

Such work requires further levels of adaptation, refinement and tailoring of a "ready" tool. Many of those requirements or good practices are not clear at the start of a project and are only arrived at gradually.

Similarly, a data sharing project needs to be integrated in a variety of forums – digital health-related, healthcare services-related, specialty-specific and ones at the intersection – that can inform its development and support its uptake. As above, many of the most relevant structures are not visible at the start of a project.

#### Acknowledge the contextual contribution

Innovators have an instinct to use opportunities which the environment opens. Perhaps they have a lesser instinct in appreciating the importance of such contextual factors for the success of their work and seeing the truly co-created nature of it.

### 3. Oppositional and/or ambivalent forces

If the category of "pure challenges" had one of its two most debated elements (generic health service challenges or IG) excluded, most of the contents of the interviews would have belonged to the third broad category of parameters, namely "oppositional or ambivalent forces".

We use "oppositional forces" to denote a relationship where a challenge is resolved through a naturally matched driver: for instance, if the challenge is lack of skills, the driver is the provision of training; if the challenge is lack of motivation, the driver is the provision of incentives. This type of relationship reflects the most typical pre-theoretical understanding of "challenges" and "drivers", which was roughly the one we also started from. As the study progressed, however, we found that in most cases where the challenges and drivers appeared pitched against one another, a driver was not simply or primarily a resolution to a challenge. The relationships between the two types of phenomena were varied and complex (see S4 File on the trajectory of conceptualising challenges and drivers in this study).

In the remainder of the Results section, we first present the two types of "oppositional forces" we formulated: 1) temporary and 2) reoccurring. We then present the most frequently discussed "ambivalent forces": 1) legitimate differences in clinical contexts or contexts requiring clinical information; 2) divergence of values or complex entanglements; and 3) unintended consequences. The other four ambivalent forces (inequality considerations; reversals of the negative; conflicting or unclear evidence; and vicarious learning and/or benefiting from the failures of others) are presented schematically in S3 File and through quotes in Box 6 and S2 File.

### 3.1. Oppositional forces, temporary

We called "oppositional forces" challenges and drivers which appeared as a negation of one another. Typically, the driver was a head-on attempt to resolve a challenge. In the case of *temporary oppositional forces*, the driver resolved or would resolve a problem. For instance, the challenge of a major upheaval within a collaborating healthcare setting, namely the introduction of an electronic patient record, was addressed by changing the intended timing of rollout. The risk of "saturation" with local end of life care projects was addressed by integrating the educational intervention of the data sharing project with a pre-existing local educational intervention in end of life care. Solutions entirely external to the project could also resolve a challenge once and for all with a single, be it complex, programme of work: e.g. the challenge of additional and multiple log-in processes, passwords and user profiles was envisaged as resolved, in the future, by a single log-in solution.

### 3.2. Oppositional forces, reoccurring

In most cases, however, the challenge-solution battle needed to be integrated into routine services or efforts needed to be renewed at regular intervals. Most prominent examples of such pairs of challenges and drivers were:

○ The lack of knowledge, skills and awareness, whether in end of life care or data sharing, vs. the provision of appropriate training initiatives.

○ The lack of information and/or the presence of negative communication patterns vs. the reliable provision of information and the engagement in good, effective communication.

○ Acting from the "wrong motives" (such as hitting organisational targets, meeting the requirements for additional financial incentives, or focusing "on what keeps the Board in a job") vs. acting in view of the patients' needs and interests.

Interviewees discussed such problems and solutions in the abstract, with only a few making a comment of their recurring nature. Even when the recurrence of the need was mentioned, it was primarily as something frustrating and inconvenient, rather than as a natural feature of certain types of work.

### 3.3. Ambivalent forces: legitimate differences in clinical contexts or contexts requiring clinical information

A very small number of interviewees (conceivably, because this was an "obvious" consideration) highlighted how different patient conditions would necessitate different choices with regard to data sharing. This concerned primarily circumstances where a patient's record contained information about, for instance, sexual health, mental illness, sexual abuse, domestic violence, criminal convictions as well as conditions which limit access to certain professions or compromise rights to insurance claims. Such cases could be identified and circumscribed for special data sharing arrangements. Yet the selective withholding of information from the whole of a patient's record was reported to be unfamiliar or difficult and to create challenges of its own.

In a further set of clinical contexts, data sharing could be largely irrelevant, e.g. in cases of seasonal illnesses or bone fractures. The presenting acute situation (over-)determined the course of action. Circumstances of extended hospital stay could also limit the value of accessing information from community services or sharing information back with them.

Most consequentially, there are irreducible differences between the record keeping needs of a health professional who generates information for their and their organisation's needs and the informational needs of other potential users. Information which enables one's own work can be irrelevant, too much, or too little for those receiving it. To address this challenge, specialist palliative care teams, for instance, were completing two templates – one for sharing and one for their internal needs, with the cross-population of information enabled in one but not in the other local specialist setting. The mismatch of needs became even more significant if the recipients were the patients themselves. While some solutions could bridge the gap, it would always remain a question of "squaring a circle".

### 3.4. Ambivalent forces, divergence or complex entanglements of values

One of the richest subtypes of "ambivalent forces" had its roots in differences of perspective, values, evaluation criteria, views, preferences and similar value-laden phenomena of which most people would argue, from a rational stance, that they vary legitimately from person to person and are a matter of personal choice. Yet most people again, when emotionally involved, would find it hard to countenance values, perspectives, views, etc. which are polar opposite to theirs without experiencing them as wrong or misguided. Such diversity and conflicts of values were relevant to core features of the EoLC data sharing project.

Most clearly, it was observed in relation to *the nature of the data sharing template* and in terms of opposing preferences for detail vs. simplicity. Some respondents argued for the benefits of the comprehensive capture of information. Others saw ultimate brevity as crucial for uptake by busy practitioners. Some preferred the template to be multifunctional, for instance including frailty and old age in addition to palliative and end of life care. Others wanted a clear end of life care scope. Some asserted the gains in shareability, monitoring and evaluation of more structured formats. Others preferred the detail and context added by accommodating more free text in the template.

Conflicts of values were observed also in relation to *the very possibility for data sharing*. Some organisations would see more benefits in keeping their in-house templates, far better

tailored to their own needs, while others would convert to templates standardised for use across the local health economy and enabling better (or even any) data sharing. The relative preference for independence, including independent faster improvement, vs. collaboration, the price for which may be slower progress, was in this case the underpinning tension of values that hinder or drive data sharing.

Conflicts of values also affected the *particular type of data sharing advanced*. For instance, some interviewees argued for the benefits of planning future care and consulting existing care plans, while others prioritised the immediate, adaptive response to the clinical situation at hand (with consequences for whether a care plan was included as a data sharing field or not). Differences of values and evaluation criteria also affected assessments of more diffuse characteristics of the data sharing project, such as its progress and success or the support structures in which it was embedded or which it had set up.

Each of the above extremes could advance some opportunities and goals (act as a driver) and constrain or block others (act as a challenge). No interviewee argued for any of the extremes, such as a tick-box only or a text-only template. Not infrequently, they acknowledged some advantages of the alternative perspective or described their own trajectory from practising differently (such as using an in-house template) to their current preference (endorsing a locality template to enable data sharing). Yet with the exception of one interviewee, who discussed pros and cons in an equitable manner and did not commit to a personal conclusion, all had clear current preferences for one or the other side on the above continua.

## 3.5. Ambivalent forces, unintended consequences

In a key pattern of a driver-challenge relationship, overall highly positive outcomes of data sharing could have harmful unintended consequences, for instance:

○ The greater ease with which GPs could obtain specialist palliative care advice, by having specialists access the GP record, could mean that more GPs felt their work was interfered with or judgement questioned.

○ The provision of more informational support to clinicians who were uncertain in their judgement could mean that they experienced greater pressure to make decisions by themselves. Alternatively, there was a risk that non-specialists became unjustifiably confident in their ability to make decisions requiring expert input.

○ Informing colleagues, through a shared record, that one had initiated an aspect of care (e.g. conversations around the end of life) could result in the other side taking such aspects of care as addressed as opposed to initiated.

○ Being able to see a well-established clinical history could prevent the fresh perspective which would have provided the solution.

○ Crucially, having up-to-date external information could be highly valuable, but having outdated information, which could be the case even if the record appeared up-to-date (e.g. because services unrepresented on the record were involved), could be far more dangerous than acting from the situation at hand.

Box 6 and Box 7 offer, respectively, illustrative quotes and recommendations associated with oppositional and ambivalent forces. S2 and S3 Files provide further detail.

Box 6. Illustrative quotes about the nine types of oppositional and ambivalent forces

### 3.1. Oppositional forces, temporary

[Name] came to talk to . . . the Education Facilitators Network . . . and I was talking to a few colleagues and we were saying, 'it's absolutely crazy that he's going to be looking to go in and be working with GP practices and yet I'm already going in to GP practices', as were they going in to trying to liaise with other services, like the community services.

I spoke to my manager and she and [names from Data Sharing Project] and myself had a meeting to discuss whether there was any mileage in trying to work together, even in the short term until the project itself had . . . taken on an education facilitator and I've sort of stayed. *[Interviewee 4]*

### 3.2. Oppositional forces, recurring

[Care homes] don't pay particularly well, and so their staff retentions are low and if you can get new staff, they leave quite often . . . So the staff just get up to scratch and we train them and we get them using, doing the DNACPR and getting engaged with it, and we get them familiar with the Just in Case bag charts and then we lose them . . . [W]e need to continue to educate the carers in the care homes because they turn over quickly. *[Interviewee 14, GP]*

### 3.3. Ambivalent forces, legitimate differences in clinical contexts or contexts requiring clinical information

[W]e will see the patient for the first time and then we have our own specialist palliative care template, which has got a lot of our national data collection type stuff on it. This is where some people have struggled as well because we're asking them to complete our own template and then to go on to complete the End of Life template. But it does have slightly different things in it, it's there for a different reason. *[Interviewee 22, palliative care professional]*

Some of the templates [of district nurses] are far too . . . do you really need to know that they were with them for 4 minutes, or it was a 4-minute entry, no, we don't, and we may well have to scroll another page and a half and read through a load of rubbish. But they obviously need it for audit and for purposes their end, so you have to just read in-between the lines sometimes. *[Interviewee 37, out of hours nurse]*

### 3.4. Ambivalent forces, divergence of values or complex entanglements

[T]he view is just incomplete. *[Interviewee 40, out of hours GP]*

vs.

I believe in very precise concise information . . . [S]ome of these templates are too long and we don't need all of that information, most of this is repetition. *[Interviewee 18, GP]*

I am very concerned. A lot of GPs are very concerned about where the NHS is heading. Obviously, we're very fiercely independent, that's why we are like we are. *[Interviewee 12, GP]*

vs.

That kind of loss of control of data is something GPs do have to accept . . . there's no way we can work individually, and I'm very happy to relinquish control to other groups or clinical teams. *[Interviewee 21, GP]*

## 3.5. Ambivalent forces, unevenness of opportunities and inequality considerations

[Data sharing] is extremely difficult, and it's extremely difficult because we work with [alternative system] and not with [dominant system], because most of the data sharing in this area is designed around [dominant system]. . . . [W]e're printing them out [care plans for vulnerable people] and then they're being uploaded as a document on [System D]. But as soon as you've done that, they're out-of-date. . . . [T]he next day the patient's medication might be changed, but that's not synchronising. So, there's actually risks with data sharing . . . very much from the practice that I work in. *[Interviewee 17, GP]*

## 3.6. Ambivalent forces, unintended consequences

Whether it was a patient, family or a specialist nurse or even a district nurse ringing up and saying, 'It's all going wrong', you could look on [System D] and see what was happening . . . and sometimes GPs were not very pleased with us for interfering. *[Interviewee 35, palliative care professional]*

[I]f we're doing the template and . . . nobody else is updating it, or nobody else is amending it, it feels like it only works for us, it doesn't give other people responsibility for managing that patient, and everybody should be responsible for managing that patient. And by not ticking the boxes that are very linked with the surgery, the red, amber, green bit, to me it felt like that at least gives me, 'well, you need to make that judgement, you as the doctor need to make that judgement'. *[Interviewee 22, palliative care professional]*

## 3.7. Ambivalent forces, reversals of the negative

Examples of the subcategory of learning from negative experiences

I have had a few patients who've declined to share notes, including a lady who certainly had a very nasty tumour.

And it was really interesting with her, because the first weekend I spoke to her, she said . . . 'I don't want you to have access to my notes and I don't want my GP to know that I've been in touch with you'. 'Fair enough, okay'.

The following weekend, she said, 'I went to my GP during the week and she hadn't got any of the information that we'd discussed'. And I said, 'Do remember that was because you declined to share information?'. She said, 'Yes'. I said, 'Well, can you now see that [it] disadvantages you?' [laughs]. And she was like, 'Oh, yeah, I understand now!'. *[Interviewee 40, out of hours GP]*

### 3.8. Ambivalent forces, conflicting or unclear evidence

I'm . . . massively in favour of it [end of life care data sharing]. It's something that is obvious, it's evidence-based that in other regions of the country, is it the South West, there's clear evidence of reduction in hospital admissions and other such benefits . . . *[Interviewee 20, palliative care professional]*

vs.

I'm not aware of anywhere else in the country that has necessarily done hugely better than us. [EPaCCS in another area], when you go to one of their presentations, you can be left thinking 'this is hugely successful', but . . . someone commented that they've got less than 1% of their population on their EPaCCS system. *[Interviewee 11]*

### 3.9. Ambivalent forces, vicarious learning or benefiting from the failures of others

[B]y and large people think it's quite a good template, certainly if you present it next to the [project name] template, everyone looks at Share my Care [original name of EoLC project] and looks at the [project name] one and thinks Share my Care is brilliant because, you know, the contrasting effect. *[Interviewee 5]*

I think I have been quite sensible and sat back and let people like [name] drive. I'm damned if I'm going to make the same effort, it would be the same thing. Let him trailblaze and then we'll follow. *[Interviewee 11]*

### Box 7. Shortlist of recommendations for action based on findings about oppositional and ambivalent forces

#### Data sharing projects are not conventional projects

Data sharing projects do not appear well suited for the traditional project format of well-defined scope, outcomes, actions, timelines, etc. They may be more adequately conceptualised and resourced as programmes of work of culture change and/or building an infrastructure.

#### The more contexts a data sharing project links, the more difficult it is to provide context-appropriate information

Consider the broad variety of contexts in which the records of a patient can be seen. To the extent to which it is possible, work to develop context-specific solutions.

Accept that some of the tensions between different types of contexts are unavoidable. Be aware that this significantly affects opportunities to develop a satisfying data sharing solution when the data sharing is across a broad range of contexts.

#### Plan for a significant diversity of perspectives and levels of power

Create structures and processes which enable everybody on the project team or stakeholder group to be heard. Pay particular attention to team members, settings, parties to the project, etc. which have (or experience themselves as having) less authority.

Create an atmosphere that allows for challenges to the dominant or senior opinions, unpalatable concerns about the project, or personal struggles around the work to be aired.

At the same time create structures, processes and standards which enable a decision to be made, even if there is significant disagreement and consensus cannot be reached.

### There will be unintended consequences

Be acutely aware of the likelihood of unintended consequences. Evaluate their presence and extent.

### (Perhaps) be gracious when winning

The success of a data sharing project tends to pass through the "corpses" of other data sharing projects. It is important to learn from the failures of others; preferable not to rejoice in them, but, realistically, cut-throat competition is part of the work on patient data sharing.

## Discussion

### Summary of main findings

Here we presented a comprehensive yet parsimonious framework of challenges, drivers, and oppositional and ambivalent forces to patient data sharing, with a focus on data sharing at the end of life. The framework was derived from a rich dataset, consisting of in-depth interviews with 44 maximum variation participants. Findings from the interviews were mapped against findings from four literature reviews on health information technology and health information exchange, covering 135 studies. Our final proposal was for a 9-component framework of "pure challenges", drawing parallels between patient data sharing and other broad and complex social and sociotechnical domains; a 4-component framework of "pure drivers", defined in terms of their internality or externality to the IT solution and project team, and thus the level of control of the latter over them; and a 9-component framework of "oppositional or ambivalent forces", reflecting the persistent tensions in challenge-driver pairs and the fact that many parameters can be simultaneously a challenge and a driver, depending on context, goals and perspective.

In addition, we derived recommendations for practical action based on or informed by the data, also building on findings from other arms of the broader study, the broader research and IT literature, and our participation in the applied aspect of the project. These are well grounded proposals but, nonetheless, require further refinement, testing and evaluation.

### Comparison with the existing literature

There is a broad range of frameworks in the field of implementation practice and science, which, in comparison to the framework proposed here, have been far more extensively articulated and researched, and widely tested in implementing innovations, whether in healthcare or other fields. Some of them have already been incorporated in this study through the inclusion of conceptualisations from the four literature reviews. In Box 8, we describe five highly influential frameworks in some detail to contextualise better the distinctiveness of the framework we are proposing.

Box 8. Examples of influential innovation implementation frameworks

THE UNITED NATIONS S.P.A.C.E. FRAMEWORK

https://un-innovation.tools/space

Clarification of tools within each module have been added if the tool headings were not perceived as self-explanatory.

### Strategy Module

**Headlines of the Future**: "Helps users to think through their innovation goals, imagine what success looks like, and discover which barriers to overcome in order to realize success."

**Scenario Blueprint**: "Helps users assess potential futures that can influence their goals and to strategize how best to prepare for these unknowns."

**Ecosystem Analysis**: "Helps users identify the actors in their innovation ecosystem and determine their comparative advantage."

**Portfolio Strategy**: "Helps users analyze an existing "portfolio" of projects focused on a specific goal, to determine whether they represent the right risk level for their innovation ambitions."

**Innovation Planner**: "Helps users plan for the execution of an individual innovation project and to identify gaps that require additional resources or additional innovative solutions."

### Partnerships Module

**Define a Value Proposition**

**Find Different Partners**

**Prepare to Partner**

**Prioritize and Select Partners**

### Architecture Module

**Scan the Horizon**: "Helps organizations source new ideas to address specific challenges."

**User-Centred Design**

**From Pilot to Scale**

**Operating Model**

### Culture Module

**Embrace Failures**

**Create Incentives and Opportunities**

**Define Strategic Risks**

**Engage Governing Bodies**

## Evaluation Module

**Innovation Storytelling**

**Stage-Gate Assessment**: "Helps users select the right methods and indicators to evaluate and make decisions about individual innovation projects."

**Life Cycle Analysis**: "Helps users identify potential bottlenecks in their processes across the innovation life cycle and develop strategies to address them."

**Enabling Environment Scan**: "Helps users survey staff perceptions of their organization's or team's culture, architecture, and partnerships."s

## THE PROSCI METHODOLOGY

www.prosci.com/methodology

**The Prosci ADKAR Model** was developed by Prosci founder Jeff Hiatt after studying the change patterns of more than 700 organizations. It is based on the understanding that organizational change can only happen when individuals change. These individuals need to be supported in their:

**Awareness** of the need to change

**Desire** to participate and support change

**Knowledge** on how to change

**Ability** to implement desired skills & behaviours

**Reinforcement** to sustain change.

The ADKAR Model is applied in combination with the Prosci 3-Phase Process and the PCT Model.

**The Prosci 3-Phase Process** focuses on the organizational change and distinguishes between:

**Phase 1: Prepare Approach** (Define Success, Define Impact, Define Approach)

**Phase 2: Manage Change** (Plan and Act, Track Performance, Adapt Actions)

**Phase 3: Sustain Outcomes** (Review Performance, Activate Sustainment, Transfer Ownership)

**The PCT (Prosci Change Triangle) model** is comprised of four aspects:

**Success** – the definition of success of the change, in terms of reasons for change, the project objectives, and organizational benefits.

**Leadership/Sponsorship**

**Project Management** – the discipline that addresses the technical side of a change by developing and delivering a solution that solves a problem or addresses an opportunity within defined scope, time and cost limits.

**Change Management** – the discipline that addresses the people side of the change.

## NASSS FRAMEWORK

Greenhalgh T, Wherton J, Papoutsi C et al. Beyond Adoption: A New Framework for Theorizing and Evaluating Nonadoption, Abandonment, and Challenges to the Scale-Up, Spread, and Sustainability of Health and Care Technologies. *J Med Internet Res* 2017; **19** (11): e367. https://www.jmir.org/2017/11/e367

The 2017 NASSS framework (a "New Framework for Theorizing and Evaluating Non-adoption, Abandonment, and Challenges to the Scale-Up, Spread and Sustainability of Health and Care Technologies") is based on a literature review of 28 technology implementation frameworks and empirical research of over 400 hours of observation, 165 interviews and 200 documents. It has seven domains:

1. The condition or illness
2. The technology
3. The value proposition (business case for developer and desirability for patients)
4. The adopter system (comprising professional staff, patient, and lay caregivers)
5. The organization(s)
6. The wider (institutional and societal) context
7. The interaction and mutual adaptation between all these domains over time.

## CONSOLIDATED FRAMEWORK FOR IMPLEMENTATION RESEARCH

Damschroder L J, Reardon C M, Widerquist M A O et al. The updated Consolidated Framework for Implementation Research based on user feedback. *Implementation Sci* 2022, **17**:75. https://doi.org/10.1186/s13012-022-01245-0

The 2022 update of the Consolidated Framework for Implementation Research (originally published in 2009, update based on a literature review of 59 articles and 134 survey responses of authors who have used the framework) is structured around 5 domains, 48 constructs and 19 subconstructs:

**Innovation domain**, with constructs of innovation Source, Evidence Base, Relative Advantage, Adaptability, Trialability, Complexity, Design and Cost.

**Outer Setting domain**, with constructs of Critical incidents, Local Attitudes, Local Conditions, Partnerships & Connections, Policies & Laws, Financing, External Pressure.

**Inner Setting domain**, with constructs of Structural Characteristics, Relational Connections, Communications, Culture, Tension for Change, Compatibility, Relative Priority, Incentive Systems, Mission Alignment, Available Resources and Access to Knowledge & Information.

**Individuals domain**, with a Roles subdomain, with constructs of High-level Leaders, Mid-level Leaders, Opinion Leaders, Implementation Facilitators, Implementation Leads, Implementation Team Members, Other Implementation Support, Innovation Delivers and Innovation Recipients; and a Characteristics subdomain, with constructs of Need, Capability, Opportunity and Motivation.

**Implementation Process**, with constructs of Teaming, Assessing Needs, Assessing Context, Planning, Tailoring Strategies, Engaging, Doing, Reflecting & Evaluating, and Adapting.

## "IMPLEMENTATION SCIENCE AT A GLANCE"

Guide of the US Department of Health & Human Services/ National Institutes of Health

https://cancercontrol.cancer.gov/sites/default/files/2020-04/NCI-ISaaG-Workbook.pdf

The guidance is structured around the phases of:

**Assess:** Engage stakeholders, Confirm evidence, Choose an intervention

**Prepare:** Maintain fidelity, Adapt intervention

**Implement**: the models given as examples to help support implementation are those of the Diffusion of innovation, Consolidated Framework for Implementation Research and Interactive Systems Framework for Dissemination and Implementation. A list of 11 categories of Implementation Strategies is also offered.

**Evaluate**, with a view to decision making about Sustainability, Scaling Up or De-Implementing.

All of the above frameworks share a focus on entities and contexts (e.g. the IT solution, the organisations, the internal and external environment), on actors and stakeholders, on phases/ timepoints, and on complex cognitive, motivational or outcomes-related constructs in need of operationalisation (e.g. knowledge, attitudes, success, value proposition, impact, performance) in addition to an emphasis on processes, strategies and actions. Our interim model (briefly noted under "Interview analysis and synthesis of interview and literature review data") followed similar conceptual lines. The framework we ultimately proposed has three main features which give it a distinctive character. This distinctiveness, we argue, justifies its application, whether as a supplementary tool or as a standalone one if further developed:

○ **Emphasis on the *form* of challenges and drivers** and a lesser concern with where they appear in terms of entity, context, or actor type (e.g. the IT solution, certain organisations, certain stakeholders); when they appear (e.g. preparation stages); what cognitive, motivational or outcomes-related construct they concern (e.g. knowledge, impact, ownership, etc.), or what activities they pertain to (e.g. designing and adapting the IT tool, resource mobilisation, or evaluating progress).

For instance, we suggest that there is value in considering "vicious circles" as one type of challenge, even if the specific vicious circles we have identified can also be considered challenges of the early stages of implementation, challenges of user attitudes, or challenges of adequate resourcing, amongst others. One of the benefits of focusing on the form here is that it alerts implementers that it is almost automatic to get trapped in a challenge unless it is addressed systematically and meticulously. Similarly, we suggest that it is helpful to see the "magnitude of repetition" as a challenge of a type, whether it concerns the repetition of setting up computer systems or the training of staff. This not only supports being realistic about resources and timelines, but also leads to expecting attrition of motivation and turnover of staff in specific tasks. Actions in response can then be proactively planned, such as job

rotations, secondments or mid-project handovers. To add an example from the drivers category, we suggest that there is value in grouping together drivers which focus on "fit", whether it is the IT solution adhering to a variety of national and local standards and requirements, or members of the project team participating in and weaving relevant stakeholder networks. Attending to a more abstract form in this case underscores the importance of also "fitting in" in a context where, typically, "standing out" is considered the road to success.

○ **Emphasis on ambivalence, ambiguity and persistent tensions** as fundamental forces in the field of innovation implementation.

Most clearly, this emphasis on ambivalence, ambiguity and persistent tensions is seen in the formulation of the category of oppositional and ambivalent forces. It is also reflected in the form of four of the nine high-level challenges (e.g. core tasks – peripheral tasks, market forces – public good) and a number of the subcategories of challenges (e.g. rationality of IT tools revealing irrationalities in the system, variety of languages, etc.).

Consistently formulating parameters with a view to perduring tensions emphasises the complexity, recalcitrance, and reoccurrence of certain challenges around data sharing and helps guard against naïve implementation planning that is suited for projects of lesser complexity. In some cases, the formulations of parameters point towards an ultimate irresolvability of a challenge, even if radical improvements are always possible. In other cases, the formulations suggest that the effective actions are not only beyond the scope of a single project, but beyond the scope of implementation work more broadly.

For instance, the "Unintended consequences" (a subtype under ambivalent forces) demonstrate some of the ways in which even the most successful form of data sharing will lead in its wake negatives that need to be controlled for. The parameters under "Technology – Human Users" ("The quality of data sharing is irreducibly determined by the record keeping practices of users" and "The quality of data sharing is irreducibly determined by the quality of classification systems and ontologies underpinning clinical IT systems", see S3 File for detail) remind us that some challenges which every data sharing project will encounter are unlikely to be successfully resolved at the level of project implementation. Low-level solutions can be offered in individual projects, such as training in adequate code assignment in patient records. Yet the long-lasting effective action is further upstream, e.g. in medical education or research on biomedical ontologies.

○ **Attention to external and longer-standing parallels as a source of fruitful learning**

Finally, all nine types of pure challenges other than the "*IT, narrowly construed*" category are shared with multiple other areas of social or sociotechnical practice, from the arrival of the railways, through working with dangerous equipment, to teaching mixed abilities classes (examples of parallel areas are offered in Box 1). Many of these areas have a long history as well as everyday presence in our lives. As such, they are potential sources of easily accessible learning, whether from ongoing implementation efforts; theoretical, empirical or historical research; or even artistic representations. If, in the context of this study, nine types of shared challenges were experienced as overwhelming, it makes sense to seek lessons outside of the field of health IT, despite the discomfort of crossing disciplinary boundaries.

## Implications for practice

Most broadly, this study underscored how important it is for data sharing project teams to look outside of the IT tools they are building and refining. Over three-thirds of the drivers and challenges we identified at the level of the 800-component intermediary framework did not pertain to the IT solution at all. While the nature of the study sample is likely to have affected

the ratio observed (users and implementers were more than IT developers), the number and magnitude of non-IT challenges and drivers identified was substantial. This seems to justify a recommendation that teams working on data sharing project need to focus persistently and intentionally on shaping the social interactions and structural and contextual parameters in the midst of which their data sharing tools are implemented, not only or primarily on perfecting them.

A high-level finding was also that challenges and drivers form a broad range of relationship patterns. The loose understanding of challenges as "negative factors" and drivers as "positive factors" that also tend to stand in simple opposition to one another is not only conceptually inaccurate (see S4 File) but also prevents us from identifying opportunities for practical action. The frequency with which complex factors can function, or appear, as both challenges and drivers highlights the vital importance of skills in eliciting the assumptions of stakeholders; in managing conflict; facilitating good decision making processes; and navigating multiple needs, interests and even worldviews, amongst others. It can be argued that "bridge-building" skills and "go-between" roles will often advance a data sharing project far more than high level IT skills and domain-specific clinical and practical knowledge.

It is also important to acknowledge that certain tensions between "a driver perspective" and "a challenge perspective" towards the same issue will never be resolved. There will always be situations when that which advances one valued goal in data sharing or patient care simultaneously blocks another valued goal. Perhaps most notably, informational needs may differ significantly across types of health professionals and contexts. It is often sorely inadequate to simply share the data. Instead, the data may need to be selected, reorganised, re-conceptualised, condensed, expanded, or otherwise enhanced so that their sharing is genuinely beneficial. This requires time and resources, both of which are in short supply in overwhelmed health systems.

More positively, organising the "pure challenges" around features and tensions familiar from other types of human endeavour points to a wide range of domains from which data sharing initiatives can draw lessons and inspiration. The IT world is unique and new (though, arguably, no longer *that* new) but it is still more akin to other forms of human endeavour than may be typically acknowledged.

## Implications for research

By now, the data of the study are dated, even if the framework we propose is at a level of generality where most issues are persistent, recurrent and will remain so for decades to come. The big picture we have drawn misses potentially crucial new elements, such as general advancement in IT and data sharing; opportunities which developments in AI may be offering; the impact of more recent legislation, such as the GDPR; new expectations for interoperability [72,73] and, crucially, the effects on digital health of the COVID-19 pandemic. Such changes may have, for instance, substantially reduced the number of "pure challenges" we formulated and this needs to be explored. The framework requires further conceptual specification, including through case studies from other clinical domains. Its practical utility also needs testing and evaluation. Whether our proposal has the potential to grow into a credible standalone implementation framework, including beyond data sharing, or can only serve to support other frameworks is, ultimately, an empirical question.

## Strengths and limitations of this study

To our knowledge, the broader study of which this sub-study is a part is the most multi-layered study on Electronic Palliative Care Coordination Systems and patient data sharing conducted

to date, combining in-depth qualitative work, embedded in findings of pre-existing literature reviews, and also involving quantitative, conceptual and critical arms reported on elsewhere. [48,58–60] The rich qualitative dataset used here and the depth of both questioning and analysis are a particular strength of this work, in a field where qualitative research is rare and of limited quality. The framework of challenges, drivers and oppositional and ambivalent forces offered is, as sought, comprehensive yet parsimonious. It was rigorously derived from the data, yet it also has a degree of conceptual discipline and coherence which many data-driven outputs lack.

This was, however, a single case study in a single locality in England, concerning a particular type of data sharing – data sharing for individual care at the end of life. Some of its findings are unlikely to be generalisable, even if we aimed to increase generalisability through including findings from literature reviews that addressed data sharing more broadly (outside of end of life care) and health information technology other than data sharing. As the depth and richness of the interview data was much higher than that of the reviews, the primary data retained a dominant role.

There are aspects of a more comprehensive picture that are missing from our work by virtue of the nature of the case study and its context, namely:

- The project relied almost fully on functionalities of pre-existing clinical IT systems. The challenges experienced in ground-breaking IT development are thus missing.

- The project has been taken forward in close collaboration with the local health economy, with many of the links with its decision makers pre-existing or with known routes to forge them. For many developers and start-ups in the context of health IT, lacking access to decision makers in the health service will be one of the biggest, if not the biggest, challenge they are experiencing.

- The UK health service – a national health system free at the point of care and with a relatively limited role of the private sector – is not a setting conducive to exploring the numerous drivers and challenges around market forces, commercialisation and competition in patient data sharing, while such factors may be leading in other countries. It is plausible, however, that they have a decisive role in a UK context too, even if insufficiently detected by this study.

The study also had both strengths and limitations in terms of sampling:

- We achieved the maximum variation we aimed for in terms of roles; contexts (including of healthcare settings and organisations but also of rurality and deprivation); clinical IT systems; types of use of the data sharing tools; background and levels of experience of health professionals; age groups, gender, conditions and type of relationship for patient and carers, etc. (see Table 1). However, we did not have as an explicit aim to achieve maximum variation in terms of aspects of Equality, Diversity and Inclusion (EDI) such as race and ethnicity, gender non-conformity, or sexual orientation. The study was set up in 2013, when such parameters of diversity could be overlooked in favour of maximising variability along other parameters.

- The narrow IT and IG specialists in the sample were six (respectively 4 and 2), even if many participants were clinicians or commissioners with notable interest in health IT and associated formal or informal roles. From the perspective of elucidating IT-specific challenges, the study may not have had enough interviewees of the right technical background. On the other hand, the maximum variation sampling approach, which meant that IT specialists were only one of the subgroups of interest, allowed for a more comprehensive picture of challenges and drivers to emerge.

As noted on several occasions and discussed in more detail under Implications for Research, the datedness of the data is also a key study weakness. The data (both interviews and literature reviews) were analysed in 2016-2017, with further conceptual analysis conducted in 2019. Since then, the paper has travelled the peer review system of several journals. It was beyond the capacity of the authors and the funding available to them to strengthen the study through the analysis of more recent data. Nonetheless, this was a rich dataset from a sensitive period in the development of data sharing solutions, complemented by literature reviews spanning publications from over two decades. Conceptual frameworks are also less prone to the detrimental effects of time and, if good enough, will both withstand new empirical tests and allow for further development and refinement.

## Conclusion

In summary, data sharing projects hold an immense promise but are hampered by challenges which, for now, appear to slow down their progress significantly and even block it completely. We hope that the framework we offered suggests clear direction for constructive action, even if some of this action will be beyond the capacity of any single data sharing team. The vision at the end of the road of significantly improved patient care, reduced duplication, costs savings and numerous other benefits, almost 60 as per our findings, is irresistible. Moreover, while challenges may be numerous and complex, no single barrier appears impossible to surmount. We have good reasons to want to chase the promise of data sharing with passion and determination.

From the perspective of this study, we are unlikely to reach it for many years to come, although in the field of innovation one should always be prepared for surprises. A further source of hope is one of the positive impacts of the mammoth challenge, and in many contexts tragedy, of the COVID-19 pandemic. It brought about the "culture change" in digital health so many were considering fundamental to its accelerated progress. We will be pleased to have been wrong in our predictions of a long and hard road ahead and observe a dramatic leap in the success of patient data sharing. In the meantime, with greater knowledge and realistic appreciation of the challenges and opportunities ahead, data sharing projects can move from an experience of "wading through treacle" to one of "making haste slowly".

## Supporting information

**S1 File. More on methods.**
(DOCX)

**S2 File. Further quotes.**
(DOCX)

**S3 File. Recommendations for action.**
(DOCX)

**S4 File. Conceptualising challenges and drivers.**
(DOCX)

## Acknowledgments

We would like to thank our interviewees, many of whom health professionals working under relentless pressures and patient and carers going through difficult end-of-life-related experiences. We would also like to thank colleagues on the service development team of the Cambridgeshire & Peterborough CCG Project for Data Sharing in End of Life Care as well as

members of the study Patient and Public Involvement Group for their support for and contribution to the project. Angela Harper and Sam Barclay provided, once again, invaluable administrative and technical help. Finally, we would like to thank our funders – the National Institute for Health Research (NIHR) Applied Research Collaboration (ARC) East of England, at Cambridgeshire and Peterborough NHS Foundation Trust, the Health Innovation and Education Cluster (HIEC) hosted by Cambridge University Health Partners (CUHP), and The Marie Curie Design to Care Programme.

## Author Contributions

**Conceptualization:** Mila Petrova, Stephen Barclay.

**Data curation:** Mila Petrova.

**Formal analysis:** Mila Petrova.

**Funding acquisition:** Stephen Barclay.

**Investigation:** Mila Petrova.

**Methodology:** Mila Petrova.

**Project administration:** Mila Petrova.

**Resources:** Stephen Barclay.

**Supervision:** Stephen Barclay.

**Validation:** Stephen Barclay.

**Writing – original draft:** Mila Petrova.

**Writing – review & editing:** Mila Petrova, Stephen Barclay.

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
