## [Decision Letter · Decision Letter 0]

26 Jul 2022

PDIG-D-21-00135

From “wading through treacle” to “making haste, slowly” in patient data sharing:

A comprehensive yet parsimonious model of drivers and challenges to data sharing based on an EPaCCS evaluation and four pre-existing literature reviews

PLOS Digital Health

Dear Dr. Petrova,

Thank you for submitting your manuscript to PLOS Digital Health. After careful consideration, we feel that it has merit but does not fully meet PLOS Digital Health's publication criteria as it currently stands. Therefore, we invite you to submit a revised version of the manuscript that addresses the points raised during the review process. In particular, both reviewers have commented on the length of the manuscript, with Reviewer #2 indicating that the current length and organization may potentially be an obstacle to comprehension. Reviewer #1 has also provided some useful suggestions for points to address in discussion.

Please submit your revised manuscript within 60 days Sep 24 2022 11:59PM. If you will need more time than this to complete your revisions, please reply to this message or contact the journal office at digitalhealth@plos.org. Please include the following items when submitting your revised manuscript:

We look forward to receiving your revised manuscript.

Kind regards,

Piyush Mathur

Section Editor

PLOS Digital Health

Journal Requirements:

Additional Editor Comments (if provided):

Reviewers' comments:

Reviewer's Responses to Questions

**Comments to the Author**

1. Does this manuscript meet PLOS Digital Health’s publication criteria? Is the manuscript technically sound, and do the data support the conclusions? The manuscript must describe methodologically and ethically rigorous research with conclusions that are appropriately drawn based on the data presented.

Reviewer #2: Partly

2. Has the statistical analysis been performed appropriately and rigorously?

Reviewer #1: Yes

Reviewer #2: N/A

3. Have the authors made all data underlying the findings in their manuscript fully available (please refer to the Data Availability Statement at the start of the manuscript PDF file)?

Reviewer #1: Yes

Reviewer #2: No

4. Is the manuscript presented in an intelligible fashion and written in standard English?

Reviewer #1: Yes

Reviewer #2: Yes

5. Review Comments to the Author

Reviewer #1: This is an interesting paper (although quite long). I do have some remarks and suggestions for improvement or further discussion:

- Data availability: I understand you cannot share data on an individual level because of privacy concerns, but are you able to share any aggregated data somehow? Perhaps text can be processed by NLP and summarized?

- Introduction: data sharing in the UK might be much different from other countries. Can you provide some introduction as to how healthcare and data sharing are setup in the UK?

- General comment: I noticed that GDPR (or the UK implementation of it) is not mentioned at once in the paper. Does GDPR influence data sharing in any way?

- General comment: There are some expressions that are perhaps clear to UK citizens but unclear to a global public. Examples: '1001 nITes', 'In the deep waters – toes in the water – high and dry', etc. (but many more!) Please explain or leave out.

- General comment: (lack of) interoperability is mentioned. Would following the FAIR principles (https://www.nature.com/articles/sdata201618) help solve some of the data sharing problems?

- 1.2.2. Fragmentation: If even in the UK (with its NHS) data there is a high level of fragmentation, how bad will this be in other countries?

- Discussion: Implications for research: there are many papers about data sharing in research, which would help to put this section in context, e.g. https://pubmed.ncbi.nlm.nih.gov/32349396/ and https://pubmed.ncbi.nlm.nih.gov/27096325/

- Line 1334-1344: As you state yourself, interviews were done nine-seven years ago. In the meantime, there have been many developments in AI, taking over human tasks. Do you see any use of AI in data sharing?

Typo:

- Line 295: "though" -> "through"

Reviewer #2: Thank you for the opportunity to review this paper. The paper aimed to identify the challenges and drivers encountered by Electronic Palliative Care Coordination Systems (EPaCCS) in comparison to those in other projects on data sharing for individual care. It aimed to propose a framework of recommendations for steps forward. The focus of this paper is important and timely within palliative care, and the manuscript contains some important messages on considerations for the implementation of EPaCCs (and other similar electronic systems) into routine practice. 

The authors are to be commended for a large in depth piece of work. However, at 13,500 words, the manuscript is too long/unmanageable to be practically useful for those interested in the topic area. While at a 'micro' level the writing is clear, the bigger picture of the paper is extremely difficult to follow. The research team have gone to considerable lengths in pulling together large amounts of data. However, the study design is confusing, particularly the way in which the qualitative analysis (primary data collection) links to the existing literature analysis. The real value of this paper lies in the primary qualitative data, with an impressive number of interviews collected, and how this informs recommendations for ways forward. The authors might consider if the paper could be improved and made more manageable by integrating findings of the literature review into the introduction / discussion. The recommendations for action (S3) are important, but felt lost in the supplementary data. The authors may also wish to consider dropping the use of popular culture metaphors and/or mnemonics; these were confusing and detracted from the key points being made. Finally, for reader to be clear on how the research was conducted, alongside make judgements of its methodological rigour, it may also be beneficial for the methods section to be integrated into the main body of the article (depending on journal styles).

6. PLOS authors have the option to publish the peer review history of their article (what does this mean?). If published, this will include your full peer review and any attached files.

**Do you want your identity to be public for this peer review?** For information about this choice, including consent withdrawal, please see our Privacy Policy.

Reviewer #1: Yes: Tim Hulsen

Reviewer #2: No

**Comments to the Author**

1. Does this manuscript meet PLOS Digital Health’s publication criteria? Is the manuscript technically sound, and do the data support the conclusions? The manuscript must describe methodologically and ethically rigorous research with conclusions that are appropriately drawn based on the data presented.

Reviewer #1: Yes

---

## [Decision Letter · Decision Letter 1]

19 Apr 2023

PDIG-D-21-00135R1

From “wading through treacle” to “making haste, slowly” in patient data sharing:

A comprehensive yet parsimonious model of drivers and challenges to data sharing based on an EPaCCS evaluation and four pre-existing literature reviews

PLOS Digital Health

Dear Dr. Petrova,

Thank you for submitting your manuscript to PLOS Digital Health. After careful consideration, we feel that it has merit but does not fully meet PLOS Digital Health's publication criteria as it currently stands. Therefore, we invite you to submit a revised version of the manuscript that addresses the points raised during the review process.

Please submit your revised manuscript within 60 days Jun 18 2023 11:59PM. If you will need more time than this to complete your revisions, please reply to this message or contact the journal office at digitalhealth@plos.org. Please include the following items when submitting your revised manuscript:

We look forward to receiving your revised manuscript.

Kind regards,

Miguel Ángel Armengol de la Hoz, Ph.D.

Section Editor

PLOS Digital Health

Journal Requirements:

1. Please send a completed 'Competing Interests' statement, including any COIs declared by your co-authors. If you have no competing interests to declare, please state "The authors have declared that no competing interests exist". Otherwise please declare all competing interests beginning with the statement "I have read the journal's policy and the authors of this manuscript have the following competing interests:"

Additional Editor Comments (if provided):

Reviewers' comments:

Reviewer's Responses to Questions

**Comments to the Author**

1. If the authors have adequately addressed your comments raised in a previous round of review and you feel that this manuscript is now acceptable for publication, you may indicate that here to bypass the “Comments to the Author” section, enter your conflict of interest statement in the “Confidential to Editor” section, and submit your "Accept" recommendation.

Reviewer #1: All comments have been addressed

Reviewer #3: All comments have been addressed

2. Does this manuscript meet PLOS Digital Health’s publication criteria? Is the manuscript technically sound, and do the data support the conclusions? The manuscript must describe methodologically and ethically rigorous research with conclusions that are appropriately drawn based on the data presented.

Reviewer #1: Yes

Reviewer #3: Partly

3. Has the statistical analysis been performed appropriately and rigorously?

Reviewer #1: Yes

Reviewer #3: N/A

4. Have the authors made all data underlying the findings in their manuscript fully available (please refer to the Data Availability Statement at the start of the manuscript PDF file)?

Reviewer #1: Yes

Reviewer #3: Yes

5. Is the manuscript presented in an intelligible fashion and written in standard English?

Reviewer #1: Yes

Reviewer #3: No

6. Review Comments to the Author

Reviewer #1: (No Response)

Reviewer #3: The manuscript is excessive long and at times difficult to read, convoluted and without focus. The subject was EPaCCS but even arguments like insufficient syringe drivers are part of the results.

-when comparing data sharing (UK concept) with Health Information Exchange, with the term more popular in the US why is not just one short paragraph ?

-when describing the NHS, why so much? the important fact is not that every big organisation is divided in sections and subsections, but the fact each one is free to choose their own software, causing interoperability issues.

-As pointed, a lot has changed since the pandemic, even EPaCCS and Respite templates. So much detail for information that can be considered too old and not reflecting fully the current environment 

-even table 1, so much information, like recruitment details, what's its purpose?

-some of the respondents comments have no interest, and they are confusing.

-1.2.3. Constant transformation, 1.2.4. Environment deprioritising “extras” and 1.3. Work in large impermanent teams are part of the same, indirect factors. Software is always evolving, as pointed, staff is always changing (A very large training institution). The focus should be on how to make progress on EPaCCS easily accessible and understood by those who could benefit from using it.

7. PLOS authors have the option to publish the peer review history of their article (what does this mean?). If published, this will include your full peer review and any attached files.

**Do you want your identity to be public for this peer review?** For information about this choice, including consent withdrawal, please see our Privacy Policy. 

Reviewer #1: Yes: Tim Hulsen

Reviewer #3: No

---

## [Decision Letter · Decision Letter 2]

13 Sep 2023

PDIG-D-21-00135R2

From “wading through treacle” to “making haste, slowly” in patient data sharing:

A comprehensive yet parsimonious model of drivers and challenges to data sharing based on an EPaCCS evaluation and four pre-existing literature reviews

PLOS Digital Health

Dear Dr. Petrova,

Thank you for submitting your manuscript to PLOS Digital Health. After careful consideration, we feel that it has merit but does not fully meet PLOS Digital Health's publication criteria as it currently stands. Therefore, we invite you to submit a revised version of the manuscript that addresses the points raised during the review process.

Please submit your revised manuscript within 60 days Nov 12 2023 11:59PM. If you will need more time than this to complete your revisions, please reply to this message or contact the journal office at digitalhealth@plos.org. Please include the following items when submitting your revised manuscript:

We look forward to receiving your revised manuscript.

Kind regards,

Miguel Ángel Armengol de la Hoz, Ph.D.

Section Editor

PLOS Digital Health

Journal Requirements:

Additional Editor Comments (if provided):

Reviewers' comments:

Reviewer's Responses to Questions

**Comments to the Author**

1. If the authors have adequately addressed your comments raised in a previous round of review and you feel that this manuscript is now acceptable for publication, you may indicate that here to bypass the “Comments to the Author” section, enter your conflict of interest statement in the “Confidential to Editor” section, and submit your "Accept" recommendation.

Reviewer #4: (No Response)

Reviewer #5: All comments have been addressed

Reviewer #6: (No Response)

Reviewer #7: (No Response)

2. Does this manuscript meet PLOS Digital Health’s publication criteria? Is the manuscript technically sound, and do the data support the conclusions? The manuscript must describe methodologically and ethically rigorous research with conclusions that are appropriately drawn based on the data presented.

Reviewer #4: Yes

Reviewer #5: Yes

Reviewer #6: No

Reviewer #7: Yes

3. Has the statistical analysis been performed appropriately and rigorously?

Reviewer #4: N/A

Reviewer #5: Yes

Reviewer #6: No

Reviewer #7: N/A

4. Have the authors made all data underlying the findings in their manuscript fully available (please refer to the Data Availability Statement at the start of the manuscript PDF file)?

Reviewer #4: Yes

Reviewer #5: No

Reviewer #6: No

Reviewer #7: Yes

5. Is the manuscript presented in an intelligible fashion and written in standard English?

Reviewer #4: Yes

Reviewer #5: Yes

Reviewer #6: No

Reviewer #7: No

6. Review Comments to the Author

Reviewer #4: Dear Dr. Petrova and Dr. Barclay,

Thank you for the opportunity to review your work. I think the paper addresses an important topic and is commendable for its work to bring organization and clarity to extremely difficult to define concepts and observations. In particular, the findings resonated with my own lived experiences in attempting to innovate healthcare systems and I do believe your research advances the field of health information exchange. My key takeaway is this paper is answering in detail what it is that makes the actual implementation and use of these highly promising tools so challenging.

While the findings in this paper are worthy of publication, I do express similar concerns as the previous reviewer. I found the paper’s writing or communication style needs to continue to be edited for clarity and further polish, particularly in the Results and Discussion section. I hope my comments and feedback below will help to support the further advancement of this paper.

Paper Title and Description – A recommendation to consider is to shift the framing of this paper from an HIE paper to be about the incentives and challenges of digital health innovation more generally (utilizing as EPaCCS as the case study/source materials). Based on the title and “Aims” statement, I expected the paper’s content to be more on the technical specifics of data-sharing itself and adjustments of practice operations to incorporate new data sources into the day-to-day workflows. As currently described in the title, this paper will likely attract the attention of technical HIE experts who are seeking that highly tactical guidance. I would consider ways to ensure the title and abstract contain key words (implementation science, project management, healthcare innovation, digital health, etc.) that will draw the attention of a broader audience of healthcare innovators and digital health leaders – those who would benefit the most from the studies findings. 

Author Summary – I find the Author Summary clear, compelling, and concise. 

Minor Revision: Please add a source to the sentence “ the evidence in their favour is uncertain.”

Introduction 

Comment: I find the Introduction clear and well-written. 

Recommendation: Consider retitling this section “Background” as oppose to “Introduction”. Per the level of detail given on HIE, NHS, and EPaCCS given, the word “Background” helps indicate to the reader that we are out of overview descriptions and getting into details. This is a personal preference though.

Minor Revision: Please add a source to statement “which are only partially supported by inconclusive and generally low-quality evidence”. 

Minor Revision: You have a wonderful aim statement written within the introduction: “We also had a strong applied goal: to derive specific and non-trivial recommendations for advancing data sharing projects, particularly ones which are in relatively early stage of their development and implementation.” I would recommend working this language into your Aims statement within the Abstract. I think it is an important goal that will help attract the readers who will most benefit from this paper. 

Minor Revision: Please relocate the paragraph labeled “Context of the broader study and service development project” to the Introduction section directly after the paragraph describing NHS. I think the description of EPaCCS better lends itself to the Introduction section, would flow nicely after the description of NHS, and creates a more organic ending to this section of the paper. 

Minor Revision: Please give 1-2 paragraphs to describe the type of patient data shared via EPaCCS and across the number and type of healthcare organizations involved (home health nursing, etc.). Also any known successes/outcomes of EPaCCS for advancing care. How much data is regularly moving across the system and/or how often is it accessed by providers? Specifically, this would be a good point to describe the template that is later referred to in the paper. Noting this information may be available in S1, the likelihood of a reader stopping to review that document is slim and such information at the outset will help to provide the additional context needed for understanding the results. 

Materials and Methods

Revision: My major concern is the 8 - 10 year timelapse of the data collected that provides the foundation of the findings for this paper. As a reader, upon learning the age of the dataset, one immediately starts to discount the potential relevance of the findings given a) the advancements in data sharing technology b) the increase user savviness with technology and b) the massive shifts in digital health fueled by the pandemic. Please include a justification (maybe in a text box cutout) to give more rational for use of this dataset. Why this dataset as oppose to conducting a new round of interviews? What indicators do you have that the dataset is still relevant to today? My suspicion is that this is an extremely high quality/hard won dataset that had more to give and deserved to be further analyzed despite its age. The reader needs more justification here than solely acknowledgement of the datasets age in order to continue reading with the confidence that there are valuable findings to come. 

Revision: You write “We acknowledge the length of time which has elapsed since data collection and have deprioritized the reporting of dated issues (e.g. claims around the need for a “culture change”, which the COVID-pandemic achieved).” Per reproducibility of this research and to build confidence in the dataset, can you provide more detail on how dated issues were identified and eliminated from the dataset?

Minor Revision: Please provide more detail on the interview time period and the stage in the implementation at which the interviews were conducted. Per the aim to support projects in their early development, it would be helpful to know if the interviewees are responding based on the initial implementation time period or several years into a fully mature project? Also, if the interviews were spread out over a long period of time, were interviewees responding based on their experiences at different points in the work. 

Minor Revision: Were all the interviews conducted in English? Do you have any demographic data of the interviewees themselves (particularly the patients/carers)? Per the increased prioritization of Diversity, Equity, and Inclusion (DEI) across healthcare, it is helpful to know if this data set represents a broad or more niche segment of healthcare. If highly niche (a healthcare organization that serves primarily high-income individuals or primarily non-English speaking patients), then I would address the need to gain perspectives from a wider variety of organizations in the limitations section of the paper as well. 

Revision: Please provide more detail on how the literature reviews were selected, the publication dates of the literature reviews themselves, and the range of publication dates of the 135 source studies reviewed. As a reader, I’m curious to know if the literature reviews cover material written in and around the interview time period or if they cover more recent literature between 2015-2023. If more recent literature is included in the literature review, I would include a note that the literature review was conducted to ensure the applicability of the interview content and further strengthen the relevancy of this paper. If the literature review is only up to 2015, I would provide justification for this decision.

Revision: Please provide the time period when the interview analysis and synthesis of interview and literature review data was conducted (page 12).

Results

Overall Comment: The results section is where the paper’s writing - at times - degrades in quality, efficiency of language, clarity, and structure. I would encourage that the author’s keep editing this section to ensure the writing is as solid as the two previous sections. Further, I would rethink the aim of this section from covering every detail uncovered in the research to instead focus on how to “teach” the reader the overview of each challenge/driver/ambivalent force. For the details, I would shift the text to supplemental documents when needed to provide immense detail on a framework component. I recommend this approach in order to make a more manageable and approachable paper for a reader to cover in one-sitting. The additional materials are there for the person who needs to study the content at its most nuanced level. 

Recommendation: I would consider listing the “pure drivers” before the “pure challenges” in the paper. This order better matches the flow of an implementation project as the incentives for conducting such work come before the challenges reveal themselves during the actual implementation. This is a personal preference though per how my brain is wired to order drivers vs. challenges. 

Revision: Given the overall length of the “results” section, I would include a short bullet point list or a table listing out the titles of each of the framework components and potential a 1-sentence description of each component. This addition will help prepare the reader for the content to come and also give them a chance to skip forward to the section of most relevance if needed. 

Revision: The section 1.1 “Radical Innovation Challenges” is a difficult read. The reader is set up for 9 challenges, 4 drivers, and 9 ambivalent forces. Then the first challenge covers 6 pages with the one challenge broken into 8 further categories further broken into further lists and individual comments. The current write-up of this first challenge is unexpected per the promise of parsimony, very weighty, and becomes immediately discouraging per the reader thinking every section is going to be broken apart this extensively (which they are not). I would encourage re-writing this section as an overview of this challenge (try to keep it to 2 pages) and then put this highly detailed text into a supplemental document dedicated to fully unwrapping this driver. 

Revision: I have a similar reaction for section 1.2. I would break these challenges down into further segments. Instead, craft an overview and put into it’s own supplemental section. 

Minor Revision: For the quotes, it’s occasionally difficult to tell where one stops and another begins. Please add a demarcation such as “Quote 1:…” to clarify. 

Minor Revision: For some quotes, you include a description of speaker’s role which is helpful for context. Please include the role for all quotes. 

Revision: While I like their inclusion, there are a lot of quotes included in the piece. I would consider continue to edit out some (limit to 2 per discussion point). My key recommendation is to target the quotes that are easiest to read both from a grammatical standpoint and don’t require significant additional context to understand. I find that some of the quotes are difficult to read without having to go over them multiple times. This makes for more frustration as a reader then benefit.

Minor Revision: Please include definitions of “internal” and “external” as they are referred to throughout the paper. Internal is referred to as the “IT solution itself”, but does that include the implementation team, the healthcare organization’s administrators, the providers, and/or extend to the patients?

Revision: I would encourage the authors to potentially bring in a copy editor to tighten up the language. At times, the writing feels too loose/first draft for an academic paper. Particularly given the length of the paper, every sentence needs to be as efficient as possible. For example on page 26:

Current version: “With minor exceptions, the discussion remained at a highly specific level – as frustration with specific people or organisations rather than as something that might be driven by the structure everybody was working in. At a most fundamental level, issues included:”

Edited version: “With minor exceptions, the discussion focused on frustrations with specific people and organizations rather than the structure of the healthcare system. Issues included:”

Revision: As the results section includes the quotes, often refers to what was “discussed”, and focusing so specifically on the EPaCCS project, it does not read as if this content also is derived from the literature reviews or a broader set of projects. I would include quotes and references to the literature reviews throughout the Discussion section. 

Recommendation: I’m curious to learn what the patients/carers thought of as key drivers and challenges to HEI given they often are a step removed from the data sharing process and/or implementation projects. Did their opinions significantly differ from the staffs? As the patient/caregiver experience is growing as an organizational priority in many healthcare centers, I would recommend a call-out box in either the results or discussion section to report on the patient/caregivers specifically. 

Discussion

Overall: I also find the discussion section to be difficult and frustrating to read. The Discussion section currently reads as if someone is speaking to me about their thoughts on the material more casually. For instance, there’s lots of extra phrases such as “Very importantly”, “for now”, “however”, “thus”. There’s sentences that read as highly complex, but I believe end up clouding the major point (example on page 61 “It is a genuine transformation of the object of study in the process of it being studied.”) There’s also strong use of the passive voice. This study is full of great material and I want the discussion section to fully serve the reader with very clear and crisp conclusions. I would recommend a strong copyediting with the goal to reduce the number of long lists, use of intermittent phrases, and simplification of sentences to help clarify the major points.

Minor revision: Add the word “were” to sentence on page 51. “The framework was derived from a rich dataset, consisting of in-depth interviews with 45 maximum variation participants, many of whom [were] “key informants”.

Revision: Another example where the writing of the paper needs editing is on page 51. 

Current version: “Findings from the interviews were also mapped against findings from four literature reviews on health information technology and health information exchange, covering 135 studies (the inclusion of one and the same study across reviews was explored and the number of unique studies is likely to be lower; yet a level of originality of interpretation is expected to characterise each review.) 

Comments: The second half of the sentence in the parathesis doesn’t make sense to me as written and also appears to be a level of detail that feels out-of-place in a summary paragraph that opens the Discussion section.

Revision: I don’t support inclusion of the paragraph on page 53 about the findings indicating that EPaCCS needs to be phased out. First, the conclusion does not fit the aim of the paper. Further, it feels like a sizable leap to assess interviews that are 8-10 years old and come to such a sizable recommendation without the consideration of more recent data. It may be a true or obvious statement for those actively working on EPaCCS, but as a reader without any further context, such a statement does not build confidence in the author’s remarks. 

Revision: On page 55, I appreciate the acknowledgement of the study representing a singular case study. Two items to address though are the literature review helped to mitigate that narrowness by bringing in content from many other studies. Could you call that out as a strength? I would also include a comment on the paper’s strength and/or limitations in regards to inclusiveness? Were the people interviewed and organizations they were affiliated with a generalizable sample of the healthcare system? If not, what other groups and/or organizations would you recommend be studied to help add to your findings? 

Recommendation: On page 56, I appreciate the write-up on limitations of the study data based on when the interviews were conducted. 

Thank you again for the opportunity to review your paper. I learned a lot from and hope my review is helpful.

Reviewer #5: This is an interesting paper which indicates the need for data sharing projects to focus less on refining their IT tools and more on shaping the relational and structural contextual parameters in the midst of which they are implemented.

Reviewer #6: Manuscript Number: PDIG-D-21-00135

Title: From “wading through treacle” to “making haste, slowly” in patient data sharing: A comprehensive yet parsimonious model of drivers and challenges to data sharing based on an EPaCCS evaluation and four pre-existing literature reviews

Short Title: Drivers and challenges to patient data sharing

This manuscript proposed a model with >500 challenges and >300 drivers for implementing Electronic Palliative Care Coordination System (EPaCCS) and other data sharing projects. Also, authors have concluded that one should focus less on refining their IT tools and more on shaping the social interactions and structural and contextual parameters for data collection.

I would like to summarize my comments in few points as:

1. Entire manuscript is written in a very subjective way or “narrative style” for data collection process, still reader will not find any glimpse of data, or any data processing methods, and outcome in table or figure.

2. The primary data source, which is 40 in-depth interviews, approximately 300,000 words is quite ambiguous and not suffice to make any conclusion.

3. In this modern era, where IoT devices are gaining enormous popularity, in my opinion (unlike the authors) one should give more importance to data collection mechanisms, devices, and IT tools for seamless, accurate and precise data collection. Also, nowadays remote, contactless methods of data collection are easier, cheaper, convenient, etc., so social interactions and other things are becoming secondary.

4. This manuscript could be written in simpler English, but abundant use of idiom, phrases, and words from classical English literature makes it hard to understand and may be not a good choice for a technical journal article.

Overall, I will not recommend this manuscript.

Reviewer #7: The authors present a framework including challenges, drivers, oppositional and and ambiguous parameters that were identified while implementing an end-of-life data sharing platform, accompanied with recommendations to move forward. Thematic analysis based on project team interviews yielded an initial framework which was used in subsequent framework analysis with remaining interviewees, using a realist approach. The authors discuss the different parameters in detail, accompanied with quotes from the interviewees. A series of practical and research-related implications are discussed. For instance, the authors propose a larger focus on social interactions and contextual parameters; as well as eliciting assumptions / perspectives from stakeholders as multiple factors can be easily seen as both drivers and challenges ("ambivalent").

Overall, I find that this paper has a great potential to contribute to both the practical implementation and academic literature on patient data sharing. It contains many interesting discussions on pros and cons, both expected and unexpected. However, I believe there are serious issues with presentation and lack of literature consideration. Although there are many issues listed, I believe that, once they are fixed, this paper has the potential to be a large contribution to the state of the art.

(1) To an extent, I agree with reviewer 3's first comment on the paper being "difficult to read". In general, the prose could be made a lot more concise. This seems to be an issue of writing style as it occurs throughout the paper. Note that many researchers are used to scanning papers for parts of interest - this is much more difficult if the text is very narrative and dense. 

It's hard to pinpoint one representative example, but I make an effort below: 

- A small example, but "The above sidelines at least two further dichotomies." One can simply say there are other dichotomies/contrasts - it makes one think why the former "sidelines" the latter (this seems irrelevant), breaking the flow of the text.

- "Expectedly, some interviewees saw an issue from the perspective of the problem" I assume the authors mean that some only look at the problems, and others tend to be more positive (?) The latter language is a lot easier to grasp.

- "One of the richest subtypes of “ambivalent forces” had its roots in differences, at times radical, of perspective, values, mindset, needs and goals, etc. relative to which a factor was considered" A factor could be seen as positive or negative depending on the perspective, thus making it ambivalent (?)

- "whether planning future care and consulting existing care plans has sufficient advantages over the immediate, adaptive response to what is, essentially, always a new clinical situation"

- "To an extent, such self-reflective comments aimed for a narrative resolution of something that might have required practical solutions"

Many of the paragraphs are quite interesting but very narrative. I often found myself looking back for the actual main points hidden in the narration, and having to write down major points to keep track.

In this vein, consider structuring Sections 1.3, 1.4 around the major concepts discussed - they are currently 1 long text, making it difficult to quickly get the main points. Similarly, for each type of ambivalent force, please summarize in the beginning how they pertain to both drivers and challenges - it currently has to be inferred from the text.

Consider structuring the implications and limitations sections using a list of bullets - I again found myself having to write down major points to keep track. A lot of very useful points are being made but are currently hidden in long narration. 

(2) The authors seem to have already dropped a number of quotes, however, the following are still unclear to me. (I am not advocating their removal but perhaps the authors can you clarify their meaning or relevance.)

- Relevance of the quote at the bottom of p. 19 is unclear.

- Quote on emergent innovation is unclear.

- Under resource limitations - some quotes here are gut wrenching, but their relevance to resource shortage for use of EoLC is not wholly clear.

- Clarify that first 4 quotes on p. 27 each separately pertain to points from bulleted list (I think??) - this was quite confusing at first.

- On p. 31, the relevance of the second part of the quote is unclear.

- The last quote on p. 38 is unclear - what does "only get one chance right at the end" mean?

(3) Note that a lot of the initial challenges pertain to a lack of user engagement, rather than purely repetiton or a vicious circle: e.g., lack of buy-in from healthcare workers (ambulance service; sticking with letters), and no time set aside for learning. I believe this should be outlined, possibly (but not necessarily) by introducing a separate parameter.

Importantly, it seems that the authors' recommendations on this topic *sidestep the entire field of change management and implementation science*. E.g., use of the ADKAR model would already help tick a lot of the challenge boxes. This greatly reduces the utility of the recommendations, which currently seem rather ad hoc.

Box 2 and 3 are more interesting, however, consider elaborating on the link between these points and discussions on drivers (e.g., this was clear to me for the "fitting in" recommendation, but not for e.g., "Data sharing projects are not projects", "be gracious when winning"). Similarly, in box 1, the part on "brutal paradox" has a sweeping statement "will thus create at least as many problems as it solves" but does not seem supported by the corresponding section.

(4) It's a bit unclear to me how the existing literature reviews were used to concretely inform the study, aside from the evaluation setup mentioning that the analysis framework was "expanded through coding the contents of the literature reviews". Please consider linking the parameters to the literature, or including this coding as supplementary information.

(5) The issues below refer to vague language being used:

- "Patient data systems are notoriously difficult to design, implement and sustain." That they certainly are - consider adding a few short reasons why that is for non-initiates.

- There is an unclear link between the EoLC data sharing project and the EPaCCS project - was the latter the target of the EoLC, or something separate (the authors mention it is only "under evaluation")? The results of the study mostly mention the EoLC project.

- "The Cambridgeshire and Peterborough EPaCCS was initiated in 2012, officially launched in 2014 and is currently “business as usual” in some respects and an evolving innovation with uncertain outcomes in others." This sentence is rather meaningless without any idea of what the former (business as usual) and the latter entail.

- "deprioritised the reporting of dated issues" - what does deprioritization mean in this setting?

- "Data collection was spread across three campaigns reflecting distinct stages" So were data collected _during_ each separate campaign?

- In table 1, what is the difference between mid-career and experienced - how are these categories defined?

- "An extensive thematic analysis was undertaken after the project team interviews" Unsure what role this refers to in table 1; project developers (those were only 6)?

- Under alternatives (challenge): "national data sharing work, such as the Summary Care Record (generic with an EoLC specific module)", "commercial products (generic, some with an EoLC-specific module)", etc. I am unsure what the parts in parenthesis mean.

- "the self-centred choices of some patients in a service free at the point of care" Could you give an example?

- The last bullet on p. 47 is unclear.

- The part on complex entanglements is unclear.

(6) Miscellaneous:

- There is a stark contrast between section 2.2, where the project team is viewed as highly positive, also by external collaborators, and section 1.3, where collaboration between team members was seen as overwhelmingly negative. I really got a mistaken impression from section 1.3 - consider nuancing this section and referring to section 2.2.

- The authors initially devised a high-order framework on a number of parameters, instead of the presented dyads. However, it is unclear how the former was more contingent on the data compared to the current approach.

- Please include short summaries for the remaining parameters (1.1.5 - 1.1.8; 3.6 - 3.9).

- "The provision of more informational support to clinicians who were uncertain in their judgement could mean that they experienced greater pressure to make decisions they felt uncomfortable with." Do the authors mean that there is a higher likelihood that decision will ultimately be wrong? Unsure where the "uncomfortable" is coming from.

7. PLOS authors have the option to publish the peer review history of their article (what does this mean?). If published, this will include your full peer review and any attached files.

**Do you want your identity to be public for this peer review?** For information about this choice, including consent withdrawal, please see our Privacy Policy. 

Reviewer #4: Yes: Casey Holmes Fee

Reviewer #5: Yes: Dr Sarah Markham

Reviewer #6: No

Reviewer #7: No

---

## [Decision Letter · Decision Letter 3]

19 Feb 2024

From “wading through treacle” to “making haste slowly”:A comprehensive yet parsimonious model of drivers and challenges to implementing patient data sharing projects based on an EPaCCS evaluation and four pre-existing literature reviews

PDIG-D-21-00135R3

Dear Dr Petrova,

We are pleased to inform you that your manuscript 'From “wading through treacle” to “making haste slowly”:A comprehensive yet parsimonious model of drivers and challenges to implementing patient data sharing projects based on an EPaCCS evaluation and four pre-existing literature reviews' has been provisionally accepted for publication in PLOS Digital Health.

Best regards,

Miguel Ángel Armengol de la Hoz, Ph.D.

Section Editor

PLOS Digital Health

Reviewer Comments (if any, and for reference):

Reviewer's Responses to Questions

**Comments to the Author**

1. If the authors have adequately addressed your comments raised in a previous round of review and you feel that this manuscript is now acceptable for publication, you may indicate that here to bypass the “Comments to the Author” section, enter your conflict of interest statement in the “Confidential to Editor” section, and submit your "Accept" recommendation.

Reviewer #5: All comments have been addressed

Reviewer #6: All comments have been addressed

2. Does this manuscript meet PLOS Digital Health’s publication criteria? Is the manuscript technically sound, and do the data support the conclusions? The manuscript must describe methodologically and ethically rigorous research with conclusions that are appropriately drawn based on the data presented.

Reviewer #5: Yes

Reviewer #6: Partly

3. Has the statistical analysis been performed appropriately and rigorously?

Reviewer #5: No

Reviewer #6: N/A

4. Have the authors made all data underlying the findings in their manuscript fully available (please refer to the Data Availability Statement at the start of the manuscript PDF file)?

Reviewer #5: Yes

Reviewer #6: Yes

5. Is the manuscript presented in an intelligible fashion and written in standard English?

Reviewer #5: Yes

Reviewer #6: Yes

6. Review Comments to the Author

Reviewer #5: On reflection I agree with other reviewers' comments and do not consider this work as demonstrating sufficient rigour to be published.

Reviewer #6: Manuscript Number: PDIG-D-21-00135R3

Title: From “wading through treacle” to “making haste slowly”:A comprehensive yet parsimonious model of drivers and challenges to implementing patient data sharing projects based on an EPaCCS evaluation and four pre-existing literature reviews

Short Title: Drivers and challenges to patient data sharing

I thank authors for considering my comments positively, answered the comments, and improved the manuscript accordingly. After revision of the manuscript, my comments are:

1. Thank you for answering my comment, and now the type of the paper is clear to me after including the paragraph – “As the case study project… …. Interested in high-level technological innovation.”

2. After clarifying the scope and conceptual nature of this work, the goal of this conceptual framework is understood. Therefore, previous comment is nullified now.

3. After adding the paragraph – “Most broadly,…… on perfecting them.” – it is clear that this work focuses on data sharing projects/ platforms rather than actually collecting the data.

4. In the revised manuscript, after following suggestions and changes made, I would recommend language of the manuscript is an important way to express the goal, methods, and results that will help attract the readers who will most benefit from this paper. Also, all quotes into boxes and make their reading more optional is a good approach.

Now, I find that this paper has a great potential to contribute on concept level of patient data sharing in an interesting narrative style.

7. PLOS authors have the option to publish the peer review history of their article (what does this mean?). If published, this will include your full peer review and any attached files.

**Do you want your identity to be public for this peer review?** For information about this choice, including consent withdrawal, please see our Privacy Policy.

Reviewer #5: **Yes: **Dr Sarah Markham

Reviewer #6: No
